# Regret Bounds and Reinforcement Learning Exploration of EXP-based Algorithms

## Abstract

We study the challenging exploration incentive problem in both bandit and reinforcement learning, where the rewards are scale-free and potentially unbounded, driven by real-world scenarios and differing from existing work. Past works in reinforcement learning either assume costly interactions with an environment or propose algorithms finding potentially low quality local maxima. Motivated by EXP-type methods that integrate multiple agents (experts) for exploration in bandits with the assumption that rewards are bounded, we propose new algorithms, namely EXP4.P and EXP4-RL for exploration in the unbounded reward case, and demonstrate their effectiveness in these new settings. Unbounded rewards introduce challenges as the regret cannot be limited by the number of trials, and selecting suboptimal arms may lead to infinite regret. Specifically, we establish EXP4.P's regret upper bounds in both bounded and unbounded linear and stochastic contextual bandits. Surprisingly, we also find that by including one sufficiently competent expert, EXP4.P can achieve global optimality in the linear case. This unbounded reward result is also applicable to a revised version of EXP3.P in the Multi-armed Bandit scenario. In EXP4-RL, we extend EXP4.P from bandit scenarios to reinforcement learning to incentivize exploration by multiple agents, including one high-performing agent, for both efficiency and excellence. This algorithm has been tested on difficult-to-explore games and shows significant improvements in exploration compared to state-of-the-art.

## 1 Introduction

Reinforcement Learning (RL) is a sequential decision-making process where a player or agent selects an action from an action space, receives the action's reward, and transitions to a new state within the state space at each time step. This process' state transitions and rewards adhere to a Markov Decision Process (MDP), represented by a transition kernel. The player's objective is to maximize the cumulative reward, which may be discounted by a parameter, by the end of the game. A significant challenge in RL involves the trade-off between exploration and exploitation. Exploration encourages the player to try new actions or arms, enhancing understanding of the game and helping future planning, albeit at the potential cost of sacrificing the immediate rewards. Conversely, exploitation focuses on maximizing the current rewards by utilizing information of known states and actions, which may prevent the player from learning more information about the game which could help to increase future rewards. To optimize cumulative rewards, the player must balance learning the game through exploration with securing immediate rewards through exploitation.

Given the existence of the state space and the dependency of actions on it, how to incentivize exploration in RL has been a central focus. A significant line of work on RL exploration leverages deep learning techniques. Utilizing deep neural networks to track $Q$-values through $Q$-networks in RL, known as DQN, demonstrates the potent synergy between deep learning and RL, as shown by (20). A simple exploration strategy based on DQN, the $\epsilon$-greedy method, was introduced in (21). Beyond $\epsilon$-greedy, intrinsic model exploration, exemplified by DORA (14) and the work of (28), calculates intrinsic rewards that directly incentivize exploration when combined with the extrinsic (actual) rewards of RL. Random Network Distillation (RND) (8), a more recent approach, depends on a fixed target network but faces risks for its local focus, lacking in global exploration efforts. Another research direction explores agent-based methods. In (16), a zero-sum Markov Game involves two players, with one aiming to maximize rewards and the other to minimize the opponent's rewards. This setup encourages the reward-maximizing player to explore more by observing its opponent's

performance. From a different angle, (33) investigates a fully decentralized homogeneous multi-agent RL setting, enabling an agent to gain global environmental insights through communication with others. Nonetheless, they all rely on each agent to communicate with the environment, and such interactions can be inefficient and executing multiple actions may incur high costs. A gap remains in how to promote exploration with fewer environment calls while still considering global environmental information, marking a significant departure addressed herein.

The exploration incentive has been garnering significant attention in Multi-armed Bandit (MAB). In MAB, the goal is to maximize the cumulative reward of a player throughout a bandit game by selecting from multiple arms at each time step, or equivalently, to minimize the regret, defined as the difference between the optimal rewards achievable and the actual obtained rewards. The contextual bandit variant enriches MAB by incorporating a context or state space $S$ and modifying the regret definition. At time step $t$, the player has context $s_t \in S$ and rewards $r^t$ follow $f(s_t)$ for a function $f$. Regret is defined by comparing the actual reward with the reward that could be achieved by the best expert, namely simple regret, or by the step-wise optimal arms, namely cumulative regret. Most existing works focus on simple regret. The contextual bandit problem has been further aligned with RL when state and reward transitions follow a MDP.

Considering this relationship, extending bandit techniques to RL is a relevant step forward. UCB (2) motivates count-based exploration (29) in RL and the subsequent Pseudo-Count exploration (5). Nevertheless, it was initially developed for stochastic bandits and imposes constraints on how the rewards are generated. General and with abundant theoretical analyses are the EXP-type MAB algorithms. Specifically, the regret of EXP3.P for adversarial bandit achieves optimality both in the expected and high probability sense. In EXP3.P, each arm has a trust coefficient (weight). The player samples each arm with probability being the sum of its normalized weights and a bias term, receives reward of the sampled arm and exponentially updates the weights based on the corresponding reward estimates. It achieves the regret of the order $O(\sqrt{T})$ in a high probability sense, though not applicable to contextual bandits with the existence of a state space. To this end, a variant of the EXP-type algorithms known as EXP4 is proposed in (3). In EXP4, there can be any number of experts. Each expert possesses a sample rule (policy) for actions (arms) and a weight. The player samples actions based on the weighted average of the experts' sample rules and updates the weights explicitly. The work on CORRAL in (1) considers a group of bandit algorithms, but it requires an implicit parameter search. EXP4 offers exploration opportunities for RL involving multiple players and one-step interactions with the environment, aspects yet to be studied. We address this gap herein as part of our contributions.

However, the existing EXP4 or its variants suffer from a strict assumption on the scale of the rewards and cannot be directly adapted to RL. It is worth noting that EXP-type algorithms are optimal under the assumption that $0 \le r_i^t \le 1$ for any arm $i$ and step $t$. The uniformly bounded assumption is crucial in the proof of regret bounds for existing EXP-type algorithms. It requires the rewards to be scalable with the knowledge of a uniform bound for all rewards in all states or context vectors. Furthermore, In the context of contextual bandit, existing methods—whether in linear contextual bandit (10), where the reward function is linear in context, or in stochastic contextual bandit (17), where both context and reward follow time-invariant distributions throughout the game—presume that rewards are bounded by 1. However, rewards in RL and contextual bandits can be unbounded and unscalable in real-world scenarios, violating the bounded assumption. Examples include navigation tasks, where the reward for each step moving the agent closer to the goal is unbounded, and racing tasks, where the reward is the distance covered by the agent. The adaptation of bandit algorithms to unbounded or scale-free cases remains unexplored. This necessitates a new algorithm based on EXP3.P and EXP4, along with a corresponding regret analysis, which motivates this paper.

Moreover, for EXP4, the expected simple regret is proven to be optimal in the contextual bandit scenario in (3). Independently, (22) proposes a modification of EXP4 that achieves a high probability guarantee, which, however, necessitates changes in the reward estimates. High probability simple regret in the original form of EXP4 has not yet been explored. Furthermore, while simple regret has been extensively studied, recent focus has shifted to cumulative regret since it characterizes global optimality, even in the stochastic contextual setting (17). Global optimality is especially important considering global exploration in RL, which has not yet been studied for EXP4, adding additional importance and relevance to our efforts.

To this end, in this paper, we are the first to propose a new algorithm, EXP4.P, based on EXP4, that does not alter the reward estimates in bandits with unbounded rewards. We demonstrate that its optimal simple regret holds with high probability and in expectation for both linear contextual bandits and stochastic contextual bandits, where the rewards may be unbounded. Extending the proof to this unbounded context is non-trivial, necessitating the application of deep results from information theory and probability. This includes establishing high-probability regret bounds in the bounded case with exponential terms and leveraging Rademacher complexity theory and sub-Gaussian properties to capture arm selection dynamics in the unbounded scenarios. Synthesizing these elements is highly technical and introduces new concepts. As a by-product, this analysis also enhances EXP3.P to yield comparable outcomes for MAB. Moreover, we also establish an upper bound on the cumulative regret in the linear case, which not only closes the existing gap, but also shows the advantage of having good enough experts for global exploration. The upper bounds for unbounded bandits necessitate a sufficiently large $T$, and we provide a worst-case analysis suggesting that no sublinear regret is attainable below a certain instance-specific minimum $T$, through our novel construction of instances.

Moreover, given the challenges in the RL context where rewards can be unbounded or unrescal-able—a situation not yet addressed by prior methods—we integrate the proposed scale-free EXP-type algorithms with deep RL. To achieve this, we extend the novel EXP4.P algorithm to RL, allowing for general experts by broadening the concept of experts to be any RL algorithms. In this framework, experts refine local policies through the underlying Markov process, and exponential weights are assigned to these experts to derive a globally optimal policy. This represents the first RL algorithm to leverage EXP-type exploration, ensuring that the overall performance is comparable to the best model even when the best model is unknown beforehand, thus facilitating model selection (19). To overcome the inefficiency of EXP4 and enable global exploration when dealing with many experts, we pair EXP4-RL with at least one state-of-the-art expert, motivated by the result on the cumulative regret in contextual bandits, enhancing both efficiency and performance. Specifically, our computational study focuses on two agents: RND and $\epsilon$-greedy DQN. We apply the EXP4-RL algorithm to challenging RL games such as Montezuma's Revenge and Mountain Car and benchmark its performance against RND (8). The empirical results demonstrate that our algorithm achieves superior exploration capabilities compared to RND by bypassing local maxima often encountered by RND. Additionally, it shows an increase in total reward as training progresses. Overall, our algorithm significantly enhances exploration in benchmark games.

While assumptions made in prior works cover several use cases, they are not applicable for emerging and upcoming cases like the ones related to RL proposed herein. For the bandit papers without i.i.d. assumptions, especially adversarial and contextual ones, the rewards are assumed to be bounded between 0 and 1, which is a very important assumption in EXP-types algorithms. By extending EXP4 to EXP4.P to unbounded sub-Gaussian settings (RL rewards are usually unbounded), we show that EXP4 and EXP3 can also work and have the same bound compared to the algorithms specifically developed for bounded settings. We also characterize the effectiveness of EXP4.P in the linear contextual bandit setting, where the rewards are not i.i.d. due to the existence of arbitrarily chosen contexts (the existing work in this context also assumes bounded rewards). We hope our discoveries in the sub-Gaussian and non-stationary contextual cases (contextual MAB is one-step RL) could motivate more work to consider powerful EXP-type algorithms in other unbounded settings, such as heavy-tailed distributions. Moreover, EXP4.P allows the use of multiple experts and can even achieve global optimality when an expert is good enough. This allows its adaptation into the context of reinforcement learning, and leads to proposed EXP4-RL.

A more thorough literature review is provided in the appendix.

The structure of the paper is as follows. In Section 2 we develop a new algorithm EXP4.P by modifying EXP4, and exhibit its regret bounds for contextual bandits and that of the EXP3.P algorithm for unbounded MAB, and lower bounds. Section 3 discusses the EXP4.P algorithm for RL exploration. Finally, in Section 4, we present numerical results related to the proposed algorithm.

## 2 REGRET BOUNDS

We first introduce notations. Let $T$ be the time horizon. For bounded bandits, at step $t$, $0 < t \leq T$ rewards $r^t$ can be chosen arbitrarily under the condition that $-1 \leq r^t \leq 1$. For unbounded bandits, let rewards $r^t$ follow multi-variate distribution $f_t(\mu, \Sigma)$ where $\mu = (\mu_1, \mu_2, \dots, \mu_K)$ is the mean vector and $\Sigma = (a_{ij})_{i,j \in \{1,\dots,K\}}$ is the covariance matrix of the $K$ arms and $f_t$ is the density.

We specify $f_t$ to be non-degenerate sub-Gaussian for analyses on light-tailed distributions where $\min_j a_{j,j} > 0$. A random variable $X$ is $\sigma^2$-sub-Gaussian if for any $t > 0$, the tail probability satisfies $P(|X| > t) \leq Be^{-\sigma^2 t^2}$ where $B$ is a positive constant.

The player receives reward $y_t = r_{a_t}^t$ by pulling arm $a_t$. The regret is defined as $R_T = \max_j \sum_{t=1}^T r_j^t - \sum_{t=1}^T y_t$ in adversarial bandits that depends on realizations of rewards. For contextual bandits with experts, besides the above let $N$ be the number of experts and $c_t$ be the context information. We denote the reward of expert $i$ by $G_i = \sum_{t=1}^T z_i(t) = \sum_{t=1}^T \xi_i(t)^T x(t)$, where $x(t) = r^t$ and $\xi_i(t) = (\xi_i^1(t), \dots, \xi_i^K(t))$ is the probability vector of expert $i$. Then regret is defined as $R_T = \max_i G_i - \sum_{t=1}^T y_t$, which is with respect to the best expert, rather than the best arm in MAB. This is reasonable since a uniform optimal arm is a special expert assigning probability 1 to the optimal arm throughout the game and experts can potentially perform better and admit higher rewards. This coincides with our generalization of EXP4.P to RL where the experts can be well-trained neural networks. We follow established definitions of pseudo regret $R_T' = T \cdot \max_k \mu_k - \sum_t E[y_t]$ and $\sum_{t=1}^T \max_i \sum_{j=1}^K \xi_i^j(t) \mu_j - \sum_t E[y_t]$ in adversarial and contextual bandits, respectively.

Meanwhile, following the existing literature, we denote $R_T^{cum}$ as the cumulative regret incorporating the contextual information. More specifically, we consider a linear contextual reward model where the reward of arm $i$ at time step $t$ is formulated as $r_i^t = c_t^T \theta_i + \delta_{i,t}$. Here $c_t$ represents the context received at time step $t$, $\theta_i$ is the time-invariant parameter unique to arm $i$, and $\delta_{i,t}$ is the noise associated with arm $i$ at time step $t$. Formally, $R_T^{cum}$ reads as $R_T^{cum} = \sum_{t=1}^T \max_i c_t^T \theta_i - \sum_{t=1}^T c_t^T \theta_{a_t}$.

Lastly, consistent with prior work, e.g., (7), we use the notation $O^*(f(t))$ for a given function $f(t)$ to represent a quantity of the order $O(f(t) \log^k f(t))$ for some integer $k$. In other words, this notation allows us to neglect the logarithmic terms when considering the order of the quantity, which is for convenience.

The core idea of the proposed algorithms herein is that by modifying the reward estimate or the weight update, we enable the characterization of EXP-type algorithms in both non-contextual and contextual settings, given unbounded reward distributions. This applies to both EXP3.P and EXP4.P, with the latter being adaptable to RL for efficient multi-expert learning. The main focus is on EXP4, as it is developed for the more general contextual setting. However, we were pleasantly surprised to find that our proof technique for EXP4.P also applies to EXP3.P, given their similar algorithmic structure with an additional term in the weight update, despite differences in the term itself. In a similar manner, we establish the result for EXP3.P, adding a valuable side contribution. Inspired by the global optimality result of EXP4.P (Theorem 4) in the linear contextual case, we extend EXP4.P to multi-expert reinforcement learning, specifically EXP4-RL, by adapting the reward estimate without a need for a large number of experts. The theoretical guarantee of EXP4-RL holds if the MDP has episodes of length 1, by fixing the running estimate $n_r$ and choosing $\Delta$ (analogous to $\alpha$ and $\gamma$ in EXP4.P) to ensure it matches the change in the reward estimate in EXP4.P.

We next demonstrate the algorithms and analyses across different scenarios.

## 2.1 Contextual Bandits and EXP4.P Algorithm

For contextual bandits, (3) give the EXP4 algorithm and prove its expected regret to be optimal under the bounded assumption on rewards and under the assumption that a uniform expert is always included, where by uniform expert we refer to an expert that always assigns equal probability to each arm. Our goal is to extend EXP4 to RL where rewards are often unbounded, such as several games in OpenAI gym, for which the theoretical guarantee of EXP4 may be absent. To this end, herein we propose a new Algorithm, named EXP4.P, as a variant of EXP4. Its effectiveness is two-fold. First, we show that EXP4.P has an optimal regret with high probability in the bounded case and consequently, we claim that the regret of EXP4.P is still optimal given unbounded bandits. All the proof are in the Appendix under the aforementioned assumption on experts. Second, it is successfully extended to RL where it achieves computational improvements.

### 2.1.1 EXP4.P Algorithm

Our proposed EXP4.P is shown as Algorithm 1. The highlighted part is the change compared to the existing EXP4 algorithm. The upper bound of the confidence interval of the reward estimate is added to the update rule for each expert, in the spirit of EXP3.P (see Algorithm 2) and removing the need of

changing the reward estimate (but quite different from that in EXP3.P for MAB). More specifically, motivated by the extension from EXP3 to EXP3.P, where an additional term in the trust coefficient (weight) is added to guarantee a high-probability regret bound, we derive the term in EXP4.P based on EXP4. To ensure a stronger result, i.e., the high-probability regret bound, this term represents another layer of exploration. If the weight of the current expert is low, meaning this expert is explored less, then the denominator is small, making the term large. This helps to increase the weight of the expert, or in other words, to explore the expert more at the next time step. Quantitatively, the value of $\alpha$ and the rate $\sqrt{NT}$ are carefully chosen to control the degree and speed of such exploration, and the choice of $\gamma$ is specifically for the uniform expert. This high-probability bound enables us to establish the regret bounds given unbounded rewards with probability $(1-\delta)(1-\eta)^T$. Subsequently, we characterize the expected regret using Rademacher complexity and VC dimension, which also apply to other algorithms.

---

**Algorithm 1** EXP4.P

---

Initialization: Weights $w_i(1) = \exp\left(\frac{\alpha\gamma}{3K}\sqrt{NT}\right)$, $i \in \{1, 2, \ldots, N\}$ where $N$ is the number of experts for $\alpha > 0$ and $\gamma \in (0, 1)$;

**for** $t = 1, 2, \ldots, T$ **do**

  The environment generates context $c_t$;

  Get probability vectors $\xi_1(t), \ldots, \xi_N(t)$ of arms from experts based on $c_t$ where $\xi_i(t) = (\xi_i^j(t))_j$;

  For any $j = 1, 2, \ldots, K$, set $p_j(t) = (1 - \gamma) \sum_{i=1}^N \frac{w_i(t) \cdot \xi_i^j(t)}{\sum_{j=1}^N w_j(t)} + \frac{\gamma}{K}$;

  Choose $i_t$ randomly according to the distribution $p_1(t), \ldots, p_K(t)$;

  Receive reward $r_{i_t}(t) = x_{i_t}(t)$;

  For any $j = 1, \ldots, K$, set $\hat{x}_j(t) = \frac{r_j(t)}{p_j(t)} \cdot \mathbb{1}_{j=i_t}$;

  Set $\hat{x}(t) = (\hat{x}_j(t))_j$;

  For any $i = 1, \ldots, N$, set

  $\hat{z}_i(t) = \xi_i(t)^T \hat{x}(t)$ and $w_i(t+1) = w_i(t) \exp\left(\frac{\gamma}{3K}\left(\hat{z}_i(t) + \frac{\alpha}{\left(\frac{w_i(t)}{\sum_{j=1}^N w_j(t)} + \frac{\gamma}{K}\right)\sqrt{NT}}\right)\right)$;

**end for**

---

### 2.1.2 BOUNDED REWARDS

Borrowing the ideas of (3), we claim EXP4.P has an optimal sublinear regret with high probability by first establishing two lemmas presented in Appendix. The main theorem is as follows. We assume that the expert family includes a uniform expert, which is also assumed in the analysis of EXP4 in (3).

**Theorem 1.** *Let $0 \leq r^t \leq 1$ for every t. For any fixed time horizon $T > 0$, for all $K$, $N \geq 2$ and for any $1 > \delta > 0$, $\gamma = \sqrt{\frac{3K \ln N}{T(\frac{2N}{3}+1)}} \leq \frac{1}{2}$, $\alpha = 2\sqrt{K \ln \frac{NT}{\delta}}$, we have that with probability at least $1 - \delta$, $R_T \leq 2\sqrt{3KT\left(\frac{2N}{3}+1\right)\ln N} + 4K\sqrt{KNT \ln\left(\frac{NT}{\delta}\right)} + 8NK \ln\left(\frac{NT}{\delta}\right)$.*

Theorem 1 implies $R_T \leq O^*(\sqrt{T})$. The regret bound does depend on $N$. In practice the number of experts is small compared to the time horizon and the independence among experts makes parallelism a possibility. Note that $\gamma < \frac{1}{2}$ for large enough $T$. The proof of Theorem 1 essentially relies on the convergence of the reward estimators, similar to that in (3). However, the objectives are different from (3), since our estimations and update of trust coefficients in EXP4.P are for experts, instead of EXP3.P for arms. This characterize the relationships among EXP4.P estimates and the actual value of experts' rewards and the total rewards gained by EXP4.P and brings non-trivial challenges.

### 2.1.3 LINEAR CONTEXTUAL BANDIT WITH UNBOUNDED REWARDS

General reward is hard to analyze due to the fact that global optimality may be intractable if the reward function is completely block-box in the given context and there are no assumptions about the distribution of contexts. To this end, some literature assumes that the contexts follow a time-invariant distribution; for example, recent work in characterizing global optimality through cumulative regret, see (17). Nevertheless, stochasticity of context can be limiting especially when considering the real-world scenarios. In a separate line of work, it is common to assume a linear reward structure, see (10). However, therein rewards are assumed to be bounded and global optimality has not yet

been studied, to the best of our knowledge. For this reason, we assume that the reward is linear in the context which reads as $r_{i,t} = c_t^T \theta_i + \delta_{i,t}$. Here $\delta_{i,t}$ follows a $\sigma_{i,t}^2$-sub-Gaussian distribution with mean 0 where $\sigma_{i,t} \leq \sigma$ and is independent across time step $t$.

**Theorem 2.** *Let context $c_t$ be chosen arbitrarily and meets the condition that $||c_t||, ||\theta_i|| \leq 1$, without loss of generality. Then we have that with probability $(1-\delta) \cdot (1 - \frac{1}{T^a})^T$ the regret of EXP4.P is $R_T \leq \log(1/\delta)O^*(\sqrt{T})$.*

Note that we do not assume any bound on $\delta_{i,t}$, unlike the prior work. The proof of Theorem 2 follows that of Theorem 3, since the rewards are still sub-Gaussian and the variance proxies are bounded by the same parameter $\sigma$. Besides this high probability regret bound, we also establish the upper bound on the pseudo regret $R_T'$ and expected regret $E[R_T]$. The formal statement reads as follows.

**Theorem 3.** *Assume the same condition as in Theorem 2. Then we have $R_T' \leq E[R_T] \leq O^*(\sqrt{T})$.*

The formal proof is deferred to Appendix; here, we present the proof logic. The proof of this theorem differs significantly from that of Theorem 6, since the rewards are no longer i.i.d. distributed. We first bound the absolute difference between $R_T$ and $R_T'$ by analyzing the non-stationary sub-Gaussian behaviors of all the rewards. Next, we decompose the expected regret $E[R_T]$ by characterizing it across different events to ensure that the value and the probability of the events cannot be too large simultaneously. In other words, either the probability of an event is small when $R_T$ is large, or the value of $R_T$ itself is small. This allows us to control $E[R_T]$ within the range of $O^*(\sqrt{T})$. Subsequently, using Jensen's inequality immediately leads to the conclusion of the first part of the inequality in the statement.

What we have established pertains to the simple regret for any policy class. Surprisingly, we obtain the following upper bound for the cumulative regret when the policy class includes an optimal policy. To the best of our knowledge, the prior work studying both simple and cumulative regret considers a stochastic contextual bandit setting (17). Our finding closes the gap in the linear contextual bandit setting under certain assumption. The formal statement reads as follows.

**Theorem 4.** *Assume the same condition as in Theorem 2. If the cumulative regret is upper bounded by $G(T)$, then the simple regret is upper bounded by $\max \{O^*(\sqrt{T}), G(T)\}$ for some function $G(T)$. Moreover, if there is a policy in the policy class $\bar{\pi} \in \{\pi_j^t\}_{1 \leq j \leq K}^{1 \leq t \leq T}$ such that $\sum_{t=1}^T \sum_{j=1}^K \bar{\pi}_j^t \mu_{j,t} \geq \sum_{t=1}^T \max_j \mu_{j,t} - F(T)$ for some function $F(T)$, then the cumulative regret of EXP4.P satisfies $R_T^{cum} \leq \max \{O^*(\sqrt{T}), F(T)\}$.*

The complete proof is in Appendix. The proof sketch is as follows. We characterize the difference between the cumulative and simple regret, and relate this difference to the gap between step-wise optimality and global optimality. The latter is determined by the performance of the policy class. With an optimal policy, we obtain the sublinear regret as stated.

The existence of such a policy is shown as follows. If the rewards are bounded, then LinUCB (10) meets the condition with $F(T) = O^*(\sqrt{T})$, $G(T) = O^*(\sqrt{T})$. If the contexts are order preserving in terms of the parameter vector $\theta$ then any optimal policy in terms of simple regret also meets the condition, since it is now also optimal in terms of cumulative regret.

This theorem demonstrates the benefits of utilizing proficient experts within the EXP4.P algorithm and fundamentally motivates extending EXP4.P to RL, building upon existing state-of-the-art methods. More specifically, it suggests that if an expert can achieve step-wise optimality (e.g., $F(T) = \sqrt{T}$), then EXP4.P can attain a similar outcome with $R_T^{cum} \leq \sqrt{T}$, enabling global exploration. Besides the theoretical statement, we also elaborate on this idea through an extensive computational study in Section 3.

### 2.1.4 STOCHASTIC CONTEXTUAL BANDIT WITH UNBOUNDED REWARDS

We proceed to show optimal regret bounds of EXP4.P for unbounded contextual bandit. Again, a uniform expert is assumed to be included in the expert family. Surprisingly, we report that the analysis can be adapted to the existing EXP3.P in next section, which leads to optimal regret in MAB under no bounded assumption which is also a new result.

**Theorem 5.** *For sub-Gaussian bandits, any time horizon $T$, for any $0 < \eta < 1$, $0 < \delta < 1$ and $\gamma, \alpha$ as in Theorem 1, with probability at least $(1-\delta)(1-\eta)^T$, EXP4.P has regret $R_T \leq$*

$$4\Delta(\eta)\Big(2\sqrt{3KT\big(\tfrac{2N}{3}+1\big)\ln N}\Big)+4\Delta(\eta)\Big(4K\sqrt{KNT\ln\big(\tfrac{NT}{\delta}\big)}+8NK\ln\big(\tfrac{NT}{\delta}\big)\Big)\ \text{where}\ \Delta(\eta)$$

is determined by $\int_{-\Delta}^{\Delta}\ldots\int_{-\Delta}^{\Delta}f\big(x_1,\ldots,x_K\big)dx_1\ldots dx_K=1-\eta$ which yields $\Delta(\eta)$ of $O(\tfrac{1}{a}\log\tfrac{1}{\eta})$.
In the proof of Theorem 5, we first perform truncation of the rewards of sub-Gaussian bandits by dividing the rewards to a bounded part and unbounded tail. For the bounded part, we directly apply the upper bound on regret of EXP4.P presented in Theorem 1 and conclude with the regret upper bound of order $O(\Delta(\eta)\sqrt{T})$. Since a sub-Gaussian distribution is a light-tailed distribution we can control the probability of the tail, i.e. the unbounded part, which leads to the overall result.

The dependence of the bound on $\Delta$ can be removed by considering large enough $T$ as stated next.
**Theorem 6.** *For sub-Gaussian bandits, for any $a>2$, $0<\delta<1$, and $\gamma,\alpha$ as in Theorem 1, EXP4.P has regret $R_T\leq\log(1/\delta)O^*(\sqrt{T})$ with probability $(1-\delta)\cdot(1-\tfrac{1}{T^a})^T$.*
Note that the constant term in $O^*(\cdot)$ depends on $a$. The above theorems deal with $R_T$; an upper bound on pseudo regret or expected regret is established next. It is easy to verify by the Jensen's inequality that $R'_T\leq E[R_T]$ and thus it suffices to obtain an upper bound on $E[R_T]$.

For bounded bandits, the upper bound for $E[R_T]$ is of the same order as $R_T$ which follows by a simple argument. For sub-Gaussian bandits, establishing an upper bound on $E[R_T]$ or $R'_T$ based on $R_T$ requires more work. We show an upper bound on $E[R_T]$ by using certain inequalities, limit theories, and Rademacher complexity. To this end, the main result reads as follows.
**Theorem 7.** *The regret of EXP4.P for sub-Gaussian bandits satisfies $R'_T\leq E[R_T]\leq O^*(\sqrt{T})$ under the assumptions stated in Theorem 6.*

## 2.2 MAB AND EXP3.P ALGORITHM

In this section, we establish upper bounds on regret in MAB given a high probability regret bound achieved by EXP3.P in (3). We revisit EXP3.P and analyze its regret in unbounded scenarios in line with EXP4.P. Formally, we show that EXP3.P achieves regret of order $O^*(\sqrt{T})$ in sub-Gaussian MAB, with respect to $R_T$, $E[R_T]$ and $R'_T$. The results are summarized as follows.

**Theorem 8.** *For sub-Gaussian MAB, any $T$, for any $0<\eta,\delta<1$, $\gamma=2\sqrt{\tfrac{3K\ln K}{5T}}$, $\alpha=2\sqrt{\ln\tfrac{NT}{\delta}}$, EXP3.P has regret $R_T\leq4\Delta(\eta)\cdot(\sqrt{KT\log(\tfrac{KT}{\delta})}+4\sqrt{\tfrac{5}{3}KT\log K}+8\log(\tfrac{KT}{\delta}))$ with probability $(1-\delta)(1-\eta)^T$ where $\Delta(\eta)=O(\tfrac{1}{a}\log\tfrac{1}{\eta})$, i.e. $\int_{-\Delta}^{\Delta}\ldots\int_{-\Delta}^{\Delta}f\big(x_1,\ldots,x_K\big)dx_1\ldots dx_K=1-\eta$.*

To proof Theorem 8, we again do truncation. We apply the bounded result of EXP3.P in (3) and achieve a regret upper bound of order $O(\Delta(\eta)\sqrt{T})$, similar to that of Theorem 5 for EXP4.P.

Similarly, we remove the dependence of the bound on $\Delta$ in Theorem 9 and claim a bound on the expected regret for sufficiently large $T$ in Theorem 10.

---
**Algorithm 2** EXP3.P

Initialization: Weights $w_i(1)=\exp\big(\tfrac{\alpha\gamma}{3}\sqrt{\tfrac{T}{K}}\big)$, $i\in\{1,2,\ldots,K\}$ for $\alpha>0$ and $\gamma\in(0,1)$;
**for** $t=1,2,\ldots,T$ **do**
    For any $i=1,2,\ldots,K$, set $p_i(t)=(1-\gamma)\tfrac{w_i(t)}{\sum_{j=1}^{K}w_j(t)}+\tfrac{\gamma}{K}$;
    Choose $i_t$ randomly according to the distribution $p_1(t),\ldots,p_K(t)$;
    Receive reward $r_{i_t}(t)$;
    For $1\leq j\leq K$, set $\hat{x}_j(t)=\tfrac{r_j(t)}{p_j(t)}\cdot\mathbb{1}_{j=i_t}$ and $w_j(t+1)=w_j(t)\exp\tfrac{\gamma}{3K}(\hat{x}_j(t)+\tfrac{\alpha}{p_j(t)\sqrt{KT}})$;
**end for**

---

**Theorem 9.** *For sub-Gaussian MAB, for $a>2$, $0<\delta<1$, and $\gamma,\alpha$ as in Theorem 8, EXP3.P has regret $R_T\leq\log(1/\delta)O^*(\sqrt{T})$ with probability $(1-\delta)\cdot(1-\tfrac{1}{T^a})^T$.*
**Theorem 10.** *The regret of EXP3.P in sub-Gaussian MAB satisfies $R'_T\leq E[R_T]\leq O^*(\sqrt{T})$ with the same assumptions as in Theorem 9.*

## 3 EXP4.P ALGORITHM FOR RL

EXP4 has shown effectiveness in contextual bandits with statistical validity. Therefore, in this section, we extend EXP4.P to RL in Algorithm 3 where rewards are assumed to be nonnegative.

The player has experts that are represented by deep $Q$-networks trained by RL algorithms (there is a one to one correspondence between the experts and $Q$-networks). Each expert also has a trust coefficient. Trust coefficients are also updated exponentially based on the reward estimates as in EXP4.P. At each step of one episode, the player samples an expert ($Q$-network) with probability that is proportional to the weighted average of expert's trust coefficients. Then $\epsilon$-greedy DQN is applied on the chosen $Q$-network. Here different from EXP4.P, the player needs to store all the interaction tuples in the experience buffer since RL is a MDP. After one episode, the player trains all $Q$-networks with the experience buffer and uses the trained networks as experts for the next episode. The basic

---

**Algorithm 3** EXP4-RL

---

Initialization: Trust coefficients $w_k = 1$ for any $k \in \{1, \ldots, E\}$, $E$ = number of experts ($Q$-networks), $K$ = number of actions, $\Delta, \epsilon, \eta > 0$ and temperature $z, \tau > 0$, $n_r = -\infty$ (an upper bound on reward);

**while** True **do**

  Initialize episode by setting $s_0$

  **for** $i = 1, 2, \ldots, T$(length of episode) **do**

    Observe state $s_i$;

    Let probability of $Q_k$-network be $\rho_k = (1 - \eta)\frac{w_k}{\sum_{j=1}^{E} w_j} + \frac{\eta}{E}$;

    Sample network $\bar{k}$ according to $\{\rho_k\}_k$;

    For $Q_{\bar{k}}$-network, use $\epsilon$-greedy to sample an action: $a^* = argmax_a Q_{\bar{k}}(s_i, a), j \in \{1, 2, \ldots, K\}, \pi_j = (1 - \epsilon) \cdot \mathbb{1}_{j=a^*} + \frac{\epsilon}{K-1} \cdot \mathbb{1}_{j \neq a^*}$;

    Sample action $a_i$ based on $\pi$;

    Interact with the environment to receive reward $r_i$ and next state $s_{i+1}$;

    $n_r = \max\{r_i, n_r\}$;

    Update the trust coefficient $w_k$ of each $Q_k$-network as follows: $P_k = \epsilon\text{-greedy}(Q_k), \hat{x}_{kj} = 1 - \frac{\mathbb{1}_{j=a^*}}{P_{kj}+\Delta}(1 - \frac{r_i}{n_r}), \forall j, y_k = E[\hat{x}_{kj}], w_k = w_k \cdot e^{\frac{y_k}{z}}$;

    Store $(s_i, a_i, r_i, s_{i+1})$ in experience replay buffer $B$;

  **end for**

  Update each expert's $Q_k$-network from buffer $B$

**end while**

---

idea is the same as in EXP4.P by using the experts that give advice vectors with deep $Q$-networks. It is a combination of deep neural networks with EXP4.P updates. From a different point of view, we can also view it as an ensemble in classification (31), by treating $Q$-networks as ensembles in RL. While general experts can be used, these are natural in a DQN framework. In our implementation and experiments we use two experts, thus $E = 2$ with two $Q$-networks. The first one is based on RND (8) while the second one is a simple DQN. To this end, in the algorithm before storing to the buffer, we also record $c_r^i = ||\hat{f}(s_i) - f(s_i)||^2$, the RND intrinsic reward as in (8). This value is then added to the 4-tuple pushed to $B$. When updating $Q_1$ corresponding to RND at the end of an iteration in the algorithm, by using $r_j + c_r^j$ we modify the $Q_1$-network and by using $c_r^j$ an update to $\hat{f}$ is executed. Network $Q_2$ pertaining to $\epsilon$-greedy is updated directly by using $r_j$. Intuitively, Algorithm 3 circumvents RND's drawback with the total exploration guided by two experts with EXP4.P updated trust coefficients. When the RND expert drives high exploration, its trust coefficient leads to a high total exploration. When it has low exploration, the second expert DQN should have a high one and it incentivizes the total exploration accordingly. Trust coefficients are updated by reward estimates iteratively as in EXP4.P, so they keep track of the long-term performance of experts and then guide the total exploration globally. These dynamics of EXP4.P combined with intrinsic rewards guarantee global exploration. The experimental results exhibited in the next section verify this intuition regarding exploration behind Algorithm 3.

We point out that potentially more general RL algorithms based on $Q$-factors can be used, e.g., boostrapped DQN (24), random prioritized DQN (23) or adaptive $\epsilon$-greedy VDBE (30) are a possibility. Furthermore, experts in EXP4 can even be policy networks trained by PPO (26) instead of DQN for exploration. A recommendation is to have a good enough expert and a small number of experts.

## 3.1 THEORETICAL RESULT

The theoretical guarantee on EXP4-RL is an implication of the current theoretical bound under certain conditions. Specifically, we have the following corollary of Theorems 4 and 7.

**Corollary** *Let us assume that the length of the Markov decision process (MDP) in RL is 1, i.e. it is reduced to a contextual bandit problem with multiple randomly drawn states, i.e., the state $s_t$ in MDP is stochastic and follows an i.i.d. distribution. Let the parameters $n_r$ and $\Delta$ be chosen to ensure that the change in the reward estimates in EXP4.P (with $\gamma = \sqrt{\frac{3K \ln N}{T(\frac{2N}{3}+1)}} \leq \frac{1}{2}$, $\alpha = 2\sqrt{K \ln \frac{NT}{\delta}}$) and EXP4-RL is equivalent. Then the results of Theorems 4 and 7, also hold, which implies that $R_T \leq O(\sqrt{T})$, where $T$ represents the number of episodes.*

**Remark** (Algorithm Consistency). *The algorithm differs from EXP4 in that the reward estimate is constructed differently, which affects how the trust coefficients are updated. If we incorporate the change in the reward estimate into the update of the trust coefficient, then this change is also reflected in the exponential term, as highlighted in EXP4.P. However, the change in this specific exponential term differs from that in EXP4.P. In other words, both are related in terms of changes in the reward estimate compared to EXP4, although there is a difference in these changes. This can be addressed by choosing the right $\alpha$ and $\gamma$ (which might be time-dependent in this case).*

## 4 COMPUTATIONAL STUDY

As a numerical demonstration of the superior performance and exploration incentive of Algorithm 3, we show the improvements on baselines on two hard-to-explore RL games, Mountain Car and Montezuma's Revenge. More precisely, we present that the real reward on Mountain Car improves significantly by Algorithm 3 in Section 4.1. Then we implement Algorithm 3 on Montezuma's Revenge and show the growing and remarkable improvement of exploration in Section 4.2. Intrinsic reward $c_r^i = ||\hat{f}(s_i) - f(s_i)||^2$ given by intrinsic model $\hat{f}$ represents the exploration of RND in (8) as introduced in Sections A and 3. We use the same criterion for evaluating exploration performance of our algorithm and RND herein. RND incentivizes local exploration with the single step intrinsic reward but with the absence of global exploration.

### 4.1 MOUNTAIN CAR

In this part, we summarize the experimental results of Algorithm 3 on Mountain Car, a classical control RL game. This game has very sparse positive rewards, which brings the necessity and hardness of exploration. Blog post (25) shows that RND based on DQN improves the performance of traditional DQN, since RND has intrinsic reward to incentivize exploration. We use RND on DQN from (25) as the baseline and show the real reward improvement of Algorithm 3, which supports the intuition and superiority of the algorithm.

The comparison between Algorithm 3 and RND is presented in Figure 1. Here the x-axis is the epoch number and the y-axis is the cumulative reward of that epoch. Figure 1a shows the raw data comparison between EXP4-RL and RND. We observe that though at first RND has several spikes exceeding those of EXP4-RL, EXP4-RL has much higher rewards than RND after 300 epochs. Overall, the relative difference of areas under the curve (AUC) is 4.9% for EXP4-RL over RND, which indicates the significant improvement of our algorithm. This improvement is better illustrated in Figure 1b with the smoothed reward values. Here there is a notable difference between EXP4-RL and RND. Note that the maximum reward hit by EXP4-RL is $-86$ and the one by RND is $-118$, which additionally demonstrates our improvement on RND. The computation complexity is in Appendix.

We conclude that Algorithm 3 performs better than the RND baseline and that the improvement increases at the later training stage. Exploration brought by Algorithm 3 gains real reward on this hard-to-explore Mountain Car, compared to the RND counterpart (without the DQN expert). The power of our algorithm can be enhanced by adopting more complex experts, not limited to only DQN.

### 4.2 MONTEZUMA'S REVENGE AND PURE EXPLORATION SETTING

In this section, we show the experimental details of Algorithm 3 on Montezuma's Revenge, another notoriously hard-to-explore RL game. The benchmark on Montezuma's Revenge is RND based on DQN which achieves a reward of zero in our environment (the PPO algorithm reported in (8) has reward 8,000 with many more computing resources; we ran the PPO-based RND with 10 parallel environments and 800 epochs to observe that the reward is also 0), which indicates that DQN has room for improvement regarding exploration.

To this end, we first implement the DQN-version RND (called simply RND hereafter) on Montezuma's Revenge as our benchmark by replacing the PPO with DQN. Then we implement Algorithm 3 with

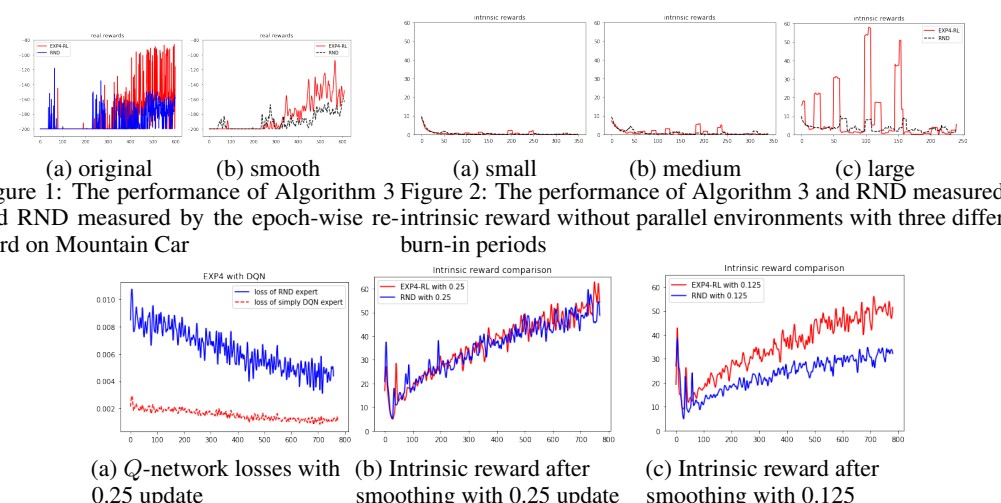

| (a) original | (b) smooth | (a) small | (b) medium | (c) large |

Figure 1: The performance of Algorithm 3 and RND measured by the epoch-wise re-intrinsic reward on Mountain Car

Figure 2: The performance of Algorithm 3 and RND measured by intrinsic reward without parallel environments with three different burn-in periods

(a) $Q$-network losses with 0.25 update    (b) Intrinsic reward after smoothing with 0.25 update    (c) Intrinsic reward after smoothing with 0.125

Figure 3: The performance of Algorithm 3 and RND with 10 parallel environments and with RND update probability 0.25 and 0.125, measured by loss and intrinsic reward.

two experts as aforementioned. Our computing environment allows at most 10 parallel environments. In subsequent figures the x-axis always corresponds to the number of epochs. RND update probability is the proportion of experience that are used for training the intrinsic model $\hat{f}$ (8).

A comparison between Algorithm 3 (EXP4-RL) and RND without parallel environments (the update probability is 100% since it is a single environment) is shown in Figure 2 with the emphasis on exploration by means of the intrinsic reward. We use 3 different numbers of burn-in periods (58, 68, 167 burn-in epochs) to remove the initial training steps, which is common in Gibbs sampling. Overall EXP4-RL outperforms RND with many significant spikes in the intrinsic rewards. The larger the number of burn-in periods is, the more significant is the dominance of EXP4-RL over RND. EXP4-RL has much higher exploration than RND at some epochs and stays close to RND at other epochs. At some epochs, EXP4-RL even has 6 times higher exploration. The relative difference in the areas under the curves are 6.9%, 17.0%, 146.0%, respectively, which quantifies the much better performance of EXP4-RL.

We next compare EXP4-RL and RND with 10 parallel environments and different RND update probabilities in Figure 3. The experiences are generated by the 10 parallel environments.

Figure 3a shows that both experts in EXP4-RL are learning with decreasing losses of their $Q$-networks. The drop is steeper for the RND expert but it starts with a higher loss. With RND update probability 0.25 in Figure 3b we observe that EXP4-RL and RND are very close when RND exhibits high exploration. When RND is at its local minima, EXP4-RL outperforms it. Usually these local minima are driven by sticking to local maxima and then training the model intensively at local maxima, typical of the RND local exploration behavior. EXP4-RL improves on RND as training progresses, e.g. the improvement after 550 epochs is higher than the one between epochs 250 and 550. In terms for AUC, this is expressed by 1.6% and 3.5%, respectively. Overall, EXP4-RL improves RND local minima of exploration, keeps high exploration of RND and induces a smoother global exploration.

With the update probability of 0.125 in Figure 3c, EXP4-RL almost always outperforms RND with a notable difference. The improvement also increases with epochs and is dramatically larger at RND's local minima. These local minima appear more frequently in training of RND, so our improvement is more significant as well as crucial. The relative AUC improvement is 49.4%. The excellent performance in Figure 3c additionally shows that EXP4-RL improves RND with global exploration by improving local minima of RND or not staying at local maxima.

Overall, with either 0.25 or 0.125, EXP4-RL incentivizes global exploration on RND by not getting stuck in local exploration maxima and outperforms RND exploration aggressively. With 0.125 the improvement with respect to RND is more significant and steady. This experimental evidence verifies our intuition behind EXP4-RL and provides excellent support for it. With experts being more advanced RL exploration algorithms, e.g. DORA, EXP4-RL can bring additional possibilities.

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

# A  LITERATURE REVIEW

The importance of exploration in RL is well understood. Count-based exploration in RL is a success story with the UCB technique. Work (29) develops the Bellman value iteration $V(s) = \max_a \hat{R}(s,a) + \gamma E[V(s')] + \beta N(s,a)^{-\frac{1}{2}}$, where $N(s,a)$ is the number of visits to $(s,a)$ for state $s$ and action $a$. Value $N(s,a)^{-\frac{1}{2}}$ is positively correlated with curiosity of $(s,a)$ and encourages exploration. This method is limited to tableau model-based MDP for small state spaces. While (5) introduces Pseudo-Count exploration for non-tabular MDP with density models, it is hard to model the concept ties to data imbalance. However, UCB achieves optimality if bandits are stochastic and may suffer linear regret otherwise (34). In the RL setting, such updates are inefficient and do not fit the dynamic RL setting. EXP-type algorithms for non-stochastic bandits can generalize to RL with fewer assumptions about the statistics of rewards, which have not yet been studied. Independently, the idea of utilizing multiple experts has been studied extensively. For example, (16) studies a zero-sum game theoretic setting and incentivizes exploration by learning from the policy and trajectory of the opponent, while (33) investigates a cooperative multi-agent learning setting where agents integrate the obtained information to make more informed decisions, with the hope of overcoming their exploitation dilemma. However, these studies all assume that different agents have varying interactions with the environment, which may be costly in the real world. In contrast, EXP-type algorithms enable multiple agents to learn from a single trajectory, necessitating our work herein. In conjunction with DQN, $\epsilon$-greedy in (21) is a simple exploration technique using DQN. Besides $\epsilon$-greedy, intrinsic model exploration computes intrinsic rewards by the accuracy of a model trained on experiences. Intrinsic rewards directly measure and incentivize exploration if added to actual rewards of RL, e.g. see (14; 28; 8). Random Network Distillation (RND) in (8) define it as $e(s',a) = \|\hat{f}(s') - f(s')\|_2^2$ where $\hat{f}$ is a parametric model and $f$ is a randomly initialized but fixed model. Here $e(s',a)$, independent of the transition, only depends on state $s'$ and drives RND to outperform others on Montezuma's Revenge. None of these algorithms use several experts which is a significant departure from our work.

Along the line of work on regret analyses focusing on EXP-type algorithms, (3) first introduces EXP3.P for bounded adversarial MAB and EXP4 for bounded contextual bandits. For the EXP3.P algorithm, an upper bound on regret of order $O(\sqrt{T})$ holds with high probability and in expectation, which has no gap with the lower bound and hence it establishes that EXP3.P is optimal. EXP4 is optimal for contextual bandits in the sense that its expected regret is $O(\sqrt{T})$. Then (22) extends it to a high probability counterpart by modifying the reward estimates. These regret bounds are invalid for bandits with unbounded support. Though (27) demonstrates a regret bound $O(\sqrt{T \cdot \gamma_T})$ for noisy Gaussian process bandits, information gain $\gamma_T$ is not well-defined in a noiseless setting. For noiseless Gaussian bandits, (15) shows both the optimal lower and upper bounds on regret, but the regret definition is not consistent with (3). Considering the more general contextual bandit, numerous analyses have focused on simple regret (6; 10), which, however, cannot uncover global optimality and thus contributes less to incentivizing global exploration. Importantly, (17) is the first not only to analyze the relationship between simple and cumulative regret but also to establish the corresponding regret upper bounds. Nevertheless, therein the context is assumed to be i.i.d. across time step $t$, specifically in a stochastic contextual bandit setting. An analysis on arbitrary contexts remains unexplored. We tackle these problems by establishing an upper bound of order $O^*(\sqrt{T})$ on regret 1) with high probability for bounded contextual bandit, 2) for linear and stochastic contextual bandit both in expectation and with high probability, and 3) for cumulative regret.

**Comparison with BEXP4 (6)** The key difference compared to BEXP4 lies in the fact that we only modify the reward estimate (resulting in a change in the weight update) following the philosophy of EXP3.P. Therefore, the modifications in our algorithm compared to EXP4 are consistent with those from EXP3 to EXP3.P, despite the values of $\gamma$ and $\alpha$ being different. However, BEXP4 modifies both the probability over actions (introducing a fixed $p_{min}$) and the reward estimate (removing adjustable $\gamma$), unlike the transition from EXP3 to EXP3.P. The necessity of this new EXP4.P lies in only modifying the reward estimate, which allows better adaptation to RL. As a by-product, its alignment with the transition from EXP3 to EXP3.P naturally extends the analysis of EXP4.P to EXP3.P. In other words, if we establish results for BEXP4, they may not work for EXP3.P and RL, which is undesirable and it establishes the importance of the proposed EXP4.P. As a result, the analysis due to the new modifications is significantly different from BEXP4, as: 1) for bounded rewards, our analysis

must be implicitly consistent with EXP3.P without experts (still challenging) while considering expert advice, and 2) for challenging unbounded rewards, we establish new analytical tools.

## B    LOWER BOUNDS ON REGRET

Algorithms can suffer extremely large regret without enough exploration when playing unbounded bandits given small $T$. To argue that our bounds on regret are not loose, we derive a lower bound on the regret for sub-Gaussian bandits that essentially suggests that no sublinear regret can be achieved if $T$ is less than an instance-dependent bound. The main technique is to construct instances that have certain regret, no matter what strategies are deployed. We need the following assumption.

**Assumption 1**    There are two types of arms with general $K$ with one type being superior ($S$ is the set of superior arms) and the other being inferior ($I$ is the set of inferior arms). Let $1 - q, q$ be the proportions of the superior and inferior arms, respectively which is known to the adversary and clearly $0 \leq q \leq 1$. The arms in $S$ are indistinguishable and so are those in $I$. The first pull of the player has two steps. First the player selects an inferior or superior set of arms based on $P(S) = 1 - q, P(I) = q$ and once a set is selected, the corresponding reward of an arm from the selected set is received.

An interesting special case of Assumption 1 is the case of two arms and $q = 1/2$. In this case, the player has no prior knowledge and in the first pull chooses an arm uniformly at random.

The lower bound is defined as $R_L(T) = \inf \sup R'_T$, where, first, $\inf$ is taken among all the strategies and then $\sup$ is among all Gaussian MAB. The following is the main result for lower bounds based on inferior arms being distributed as $\mathcal{N}(0, 1)$ and superior as $\mathcal{N}(\mu, 1)$ with $\mu > 0$.

**Theorem 11.** *In Gaussian MAB under Assumption 1, for any $q \geq 1/3$ we have $R_L(T) \geq (q - \epsilon) \cdot \mu \cdot T$, where $\mu$ has to satisfy $G(q, \mu) < q$ with $\epsilon$ and $T$ determined by $G(q, \mu) < \epsilon < q, T \leq$*

$$\frac{\epsilon - G(q, \mu)}{(1-q) \cdot \int \left| e^{-\frac{x^2}{2}} - e^{-\frac{(x-\mu)^2}{2}} \right|} + 2 \text{ where } G(q, \mu) \text{ is } \max\{ \int |q e^{-\frac{x^2}{2}} - (1 - q) e^{-\frac{(x-\mu)^2}{2}}| dx,$$

$$\int |(1 - q) e^{-\frac{x^2}{2}} - q e^{-\frac{(x-\mu)^2}{2}}| dx \}.$$

To prove Theorem 11, we construct a special subset of Gaussian MAB with equal variances and zero covariances. On these instances we find a unique way to explicitly represent any policy. This builds a connection between abstract policies and this concrete mathematical representation. Then we show that pseudo regret $R'_T$ must be greater than certain values no matter what policies are deployed, which indicates a regret lower bound on this subset of instances.

Feasibility of the aforementioned conditions is established in the following theorem.

**Theorem 12.** *In Gaussian MAB under Assumption 1, for any $q \geq 1/3$, there exist $\mu$ and $\epsilon, \epsilon < \mu$ such that $R_L(T) \geq (q - \epsilon) \cdot \mu \cdot T$.*

The following result with two arms and equal probability in the first pull deals with general MAB. It shows that for any fixed $\mu > 0$ there is a minimum $T$ and instances of MAB so that no algorithm can achieve sublinear regret. Table 1 (see Appendix) exhibits how the threshold of $T$ varies with $\mu$.

**Theorem 13.** *For general MAB under Assumption 1 with $K = 2, q = 1/2$, we have that $R_L(T) \geq \frac{T \cdot \mu}{4}$ holds for any distributions $f_0$ for the arms in $I$ and $f_1$ for the arms in $S$ with $\int |f_1 - f_0| > 0$ (possibly with unbounded support), for any $\mu > 0$ and $T$ satisfying $T \leq \frac{1}{2 \cdot \int |f_0 - f_1|} + 1$.*

## C    DETAILS ABOUT NUMERICAL EXPERIMENTS

### C.1    MOUNTAIN CAR

For the Mountain Car experiment, we use the Adam optimizer with the $2 \cdot 10^{-4}$ learning rate. The batch size for updating models is 64 with the replay buffer size of 10,000. The remaining parameters are as follows: the discount factor for the $Q$-networks is 0.95, the temperature parameter $\tau$ is 0.1, $\eta$ is 0.05, and $\epsilon$ is decaying exponentially with respect to the number of steps with maximum 0.9 and minimum 0.05. The length of one epoch is 200 steps. The target networks load the weights and biases of the trained networks every 400 steps. Since a reward upper bound is known in advance, we use $n_r = 1$.

We next introduce the structure of neural networks that are used in the experiment. The neural networks of both experts are linear. For the RND expert, it has the input layer with 2 input neurons,

followed by a hidden layer with 64 neurons, and then a two-headed output layer. The first output layer represents the $Q$ values with 64 hidden neurons as input and the number of actions output neurons, while the second output layer corresponds to the intrinsic values, with 1 output neuron. For the DQN expert, the only difference lies in the absence of the second output layer.

**Computational complexity** On Mountain Car, the runtime of EXP4-RL is about 13 hours, while the runtime of RND is about 10 hours. This implies the efficiency of the proposed algorithm since the total operation time of an iteration is approximately determined by the sum of the operation times the number of experts. RND runs much slower compared to DQN as it maintains more complex neural networks. By enabling the use of sufficiently good experts, we eliminate the need for a large number of experts, addressing the bottleneck of adapting EXP-type algorithms to RL. For EXP4.P and EXP3.P, the total operation time is the same as for EXP4 and EXP3, respectively, noting that the changes are in the construction of the reward estimates or, equivalently, the construction of the trust coefficients.

### C.2 Montezuma's Revenge

For the Montezuma's Revenge experiment, we use the Adam optimizer with the $10^{-5}$ learning rate. The other parameters read: the mini batch size is 4, replay buffer size is 1,000, the discount factor for the $Q$-networks is 0.999 and the same valus is used for the intrinsic value head, the temperature parameter $\tau$ is 0.1, $\eta$ is 0.05, and $\epsilon$ is increasing exponentially with minimum 0.05 and maximum 0.9. The length of one epoch is 100 steps. Target networks are updated every 300 steps. Pre-normalization is 50 epochs and the weights for intrinsic and extrinsic values in the first network are 1 and 2, respectively. The upper bound on reward is set to be constant $n_r = 1$.

For the structure of neural networks, we use CNN architectures since we are dealing with videos. More precisely, for the $Q$-network of the DQN expert in EXP4-RL and the predictor network $\hat{f}$ for computing the intrinsic rewards, we use Alexnet (18) pretrained on ImageNet (11). The number of output neurons of the final layer is 18, the number of actions in Montezuma. For the RND baseline and RND expert in EXP4-RL, we customize the $Q$-network with different linear layers while keeping all the layers except the final layer of pretrained Alexnet. Here we have two final linear layers representing two value heads, the extrinsic value head and the intrinsic value head. The number of output neurons in the first value head is again 18, while the second value head is with 1 output neuron.

More details about the setup of the experiment on Montezuma's Revenge are elaborated as follows. The experiment of RND with PPO in **(author?)** (8) uses many more resources, such as 1024 parallel environments and runs 30,000 epochs for each environment. Parallel environments generate experiences simultaneously and store them in the replay buffer. Our computing environment allows at most 10 parallel environments. For the DQN-version of RND, we use the same settings as **(author?)** (8), such as observation normalization, intrinsic reward normalization and random initialization. RND update probability is the proportion of experience in the replay buffer that are used for training the intrinsic model $\hat{f}$ in RND (8). Here in our experiment, we compare the performance under 0.125 and 0.25 RND update probability.

## D PROOF OF RESULTS IN SECTION 3.1

We first present two lemmas that characterize the relationships among our EXP4.P estimations, the true rewards, and the reward gained by EXP4.P, building on which we establish an optimal sublinear regret of EXP4.P with high probability in the bounded case.

The estimated reward of expert $i$ and the gained reward by the EXP4.P algorithm is denoted by $\hat{G}_i = \sum_{t=1}^{T} \hat{z}_i(t)$ and $G_{EXP4.P} = \sum_{t=1}^{T} y(t) = \sum_{t=1}^{T} r_{i_t}^t$, respectively.

For simplicity, we denote

$$\hat{\sigma}_i(t+1) = \sqrt{NT} + \sum_{l=1}^{t} \left( \frac{1}{\left( \frac{w_i(l)}{\sum_j w_j(l)} + \frac{\gamma}{K} \right) \cdot \sqrt{NT}} \right)$$

$$U = \max_i (\hat{G}_i + \alpha \cdot \hat{\sigma}_i(T+1)), \quad q_i(t) = \frac{w_i(t)}{\sum_j w_j(t)}.$$

Let $\alpha$ be the parameter specified in Algorithm 3. The lemmas read as follows.

**Lemma 1.** *If* $2\sqrt{K \ln \frac{NT}{\delta}} \leq \alpha \leq 2\sqrt{NT}$ *and* $\gamma < \frac{1}{2}$, *then* $P(\exists \, i, \hat{G}_i + \alpha \cdot \hat{\sigma}_i(T+1) < G_i) \leq \delta$.

**Lemma 2.** *If* $\alpha \leq 2\sqrt{NT}$, *then* $G_{EXP4.P} \geq \left(1 - (1 + \frac{2N}{3})\gamma\right) \cdot U - \frac{3K}{\gamma} \ln N - 2\alpha K\sqrt{NT} - 2\alpha^2$.

### D.1 Proof of Lemma 1

*Proof.* Let us denote $s_t = \frac{\alpha}{2\hat{\sigma}_i(t+1)}$. Since $\alpha \leq 2\sqrt{NT}$ by assumption and $\hat{\sigma}_i(t+1) \geq \sqrt{NT}$ by its definition, we have that $s_t \leq 1$. Meanwhile,

$$P\left(\hat{G}_i + \alpha\hat{\sigma}_i(T+1) < G_i\right)$$

$$= P\left(\sum_{t=1}^{T}(z_i(t) - \hat{z}_i(t)) - \frac{\alpha\hat{\sigma}_i(T+1)}{2} > \frac{\alpha\hat{\sigma}_i(T+1)}{2}\right)$$

$$\leq P\left(\frac{1}{K}s_T\sum_{t=1}^{T}\left(z_i(t) - \hat{z}_i(t) - \frac{\alpha}{2\left(q_i(t) + \frac{\gamma}{K}\right)\sqrt{NT}}\right) > \frac{\alpha^2}{4K}\right)$$

$$\leq e^{-\frac{\alpha^2}{4K}} E\left[\exp\left(\frac{s_T}{K}\sum_{t=1}^{T}\left(z_i(t) - \hat{z}_i(t) - \frac{\alpha}{2\left(q_i(t) + \frac{\gamma}{K}\right)\sqrt{NT}}\right)\right)\right] \quad (1)$$

where the first inequality holds by multiplying $\frac{1}{K}s_T = \frac{1}{K} \cdot \frac{\alpha}{2\hat{\sigma}_i(T+1)}$ on both sides and then using the fact that $\hat{\sigma}_i(T+1) > \sum_{t=1}^{T}\left(\frac{1}{\left(q_i(t) + \frac{\gamma}{K}\right) \cdot \sqrt{NT}}\right)$ and the second one holds by the Markov's inequality.

We introduce variable $V_t = \exp\left(\frac{s_t}{K}\sum_{t'=1}^{t}\left(z_i(t') - \hat{z}_i(t') - \frac{\alpha}{2\left(q_i(t') + \frac{\gamma}{K}\right)\sqrt{NT}}\right)\right)$ for any $t = 1, \ldots, T$. Probability (1) can be expressed as $e^{-\frac{\alpha^2}{4K}} E[V_T]$. We denote $\mathcal{F}_{t-1}$ as the filtration of the past $t - 1$ observations. Note that $V_t = \exp\left(\frac{s_t}{K}\left(z_i(t) - \hat{z}_i(t) - \frac{\alpha}{2\left(q_i(t) + \frac{\gamma}{K}\right)\sqrt{NT}}\right)\right) \cdot V_{t-1}^{\frac{s_t}{s_{t-1}}}$ and $s_t$ is deterministic given $\mathcal{F}_{t-1}$ since it depends on $q_i(\tau)$ up to time $t$ and $q_i(t)$ is computed by the past $t - 1$ rewards.

Therefore, we have

$$E[V_t|\mathcal{F}_{t-1}]$$

$$= E\left[\exp\left(\frac{s_t}{K}\left(z_i(t) - \hat{z}_i(t) - \frac{\alpha}{2\left(q_i(t) + \frac{\gamma}{K}\right)\sqrt{NT}}\right)\right) \cdot (V_{t-1})^{\frac{s_t}{s_{t-1}}} |\mathcal{F}_{t-1}\right]$$

$$= E_t\left[\exp\left(\frac{s_t}{K}\left(z_i(t) - \hat{z}_i(t) - \frac{\alpha}{2\left(q_i(t) + \frac{\gamma}{K}\right)\sqrt{NT}}\right)\right)\right] \cdot (V_{t-1})^{\frac{s_t}{s_{t-1}}}$$

$$\leq E_t\left[\exp\left(\frac{s_t}{K}\left(z_i(t) - \hat{z}_i(t) - \frac{s_t}{q_i(t) + \frac{\gamma}{K}}\right)\right) \cdot (V_{t-1})^{\frac{s_t}{s_{t-1}}}\right]$$

$$\leq E_t\left[1 + \frac{s_t(z_i(t) - \hat{z}_i(t))}{K} + \frac{s_t^2(z_i(t) - \hat{z}_i(t))^2}{K^2}\right]\exp\left(-\frac{s_t^2}{K\left(q_i(t) + \frac{\gamma}{K}\right)}\right) \cdot (V_{t-1})^{\frac{s_t}{s_{t-1}}} \quad (2)$$

where the first inequality holds by using the fact that

$$\alpha = 2s_t \cdot \hat{\sigma}_i(t+1) \geq 2s_t \cdot \sqrt{NT} \geq s_t \cdot \sqrt{NT}$$

by its definition and the second inequality holds since $e^x \leq 1 + x + x^2$ for $x < 1$ which is guaranteed by $s_t < 1$ and $z_i(t) - \hat{z}_i(t) < 1$. The latter one holds by $1 \geq r > 0$ and $x_i(t) - \hat{x}_i(t) = (1 - \frac{1}{p})x_{i_t}(t) \leq 1$

for $x_{i_t}(t) > 0$ and

$$
\begin{aligned}
z_i(t) - \hat{z}_i(t) &= \sum_{j=1}^{K} \xi_i^j(t)(x_i(t) - \hat{x}_i(t)) \\
&= \sum_{j \neq i_t} \xi_i^j(t) x_j(t) + \xi_i^{i_t}(t)(x_i(t) - \hat{x}_i(t)) \\
&\leq \sum_{j \neq i_t} \xi_i^j(t) + \xi_i^{i_t}(t) \\
&= \sum_{j=1}^{K} \xi_i^j(t) = 1
\end{aligned}
$$

Meanwhile,

$$
E[\hat{z}_i(t)] = E\left[\sum_{j=1}^{K} \xi_i^j(t)\hat{x}_j(t)\right] = \sum_{j=1}^{K} \xi_i^j(t) E[\hat{x}_j(t)] = \sum_{j=1}^{K} \xi_i^j(t) \cdot x_j(t) = z_i(t)
$$

and

$$
\begin{aligned}
&E\left[(\hat{z}_i(t) - z_i(t))^2\right] \\
&= E\left[\left(\sum_{j=1}^{K} \xi_i^j(t)\left(\hat{x}_j(t) - x_j(t)\right)\right)^2\right] \\
&\leq K \sum_{j=1}^{K} \left(\xi_i^j(t)\right)^2 E\left[(\hat{x}_j(t) - x_j(t))^2)\right] \\
&= K \sum_{j=1}^{K} \left(\xi_i^j(t)\right)^2 \cdot \left(\frac{p_j(t)\left(x_j(t)\right)^2\left(1 - p_j(t)\right)}{p_j(t)} + \left(1 - p_j(t)\right)\left(x_j(t)\right)^2\right) \\
&= K \sum_{j=1}^{K} \left(\xi_i^j(t)\right)^2 \cdot 2(1 - p_j(t))\left(x_j(t)\right)^2 \\
&\leq K \sum_{j=1}^{K} \left(\xi_i^j(t)\right)^2 \cdot \frac{1 - \gamma}{p_j(t)}
\end{aligned}
$$

where the first inequality holds by the Cauchy Schwarz inequality and the second inequality holds by the fact that $x_j(t) \leq 1$ and $2(1 - p)p < 1 - \gamma$ since $\gamma < \frac{1}{2}$ by assumption.

Note that for any $i, j = 1, \ldots, N$ we have

$$
\begin{aligned}
p_j(t) &= (1 - \gamma)\sum_{\bar{i}=1}^{N} q_{\bar{i}}(t) \cdot \xi_{\bar{i}}^j(t) + \frac{\gamma}{K} \\
&\geq (1 - \gamma)q_i(t) \cdot \xi_i^j(t) + (1 - \gamma)\frac{\gamma}{K} \cdot \xi_i^j(t) \\
&= (1 - \gamma)(q_i(t) + \frac{\gamma}{K}) \cdot \xi_i^j(t).
\end{aligned}
$$

since $1 - \gamma \leq 1, \xi_i^j(t) \leq 1$.

We further bound

$$E\left[(\hat{z}_i(t) - z_i(t))^2\right]$$

$$\leq K\sum_{j=1}^{K}\left(\xi_i^j(t)\right)^2 \cdot \frac{1-\gamma}{p_j(t)}$$

$$\leq K\sum_{j=1}^{K}\frac{(1-\gamma)\left(\xi_i^j(t)\right)^2}{(1-\gamma)(q_i(t)+\frac{\gamma}{K})\xi_i^j(t)}$$

$$= K\sum_{j=1}^{K}\frac{\xi_i^j(t)}{q_i(t)+\frac{\gamma}{K}}$$

$$= K\frac{1}{q_i(t)+\frac{\gamma}{K}}$$

Then by using (2), we have that

$$E[V_t|\mathcal{F}_{t-1}] \leq \left(1+\frac{s_t^2}{K\left(q_i(t)+\frac{\gamma}{K}\right)}\right)\exp\left(-\frac{s_t^2}{K\left(q_i(t)+\frac{\gamma}{K}\right)}\right)\cdot (V_{t-1})^{\frac{s_t}{s_{t-1}}}$$

$$\leq \exp\left(\frac{s_t^2}{K\left(q_i(t)+\frac{\gamma}{K}\right)}-\frac{s_t^2}{K\left(q_i(t)+\frac{\gamma}{K}\right)}\right)(V_{t-1})^{\frac{s_t}{s_{t-1}}}$$

$$\leq 1+V_{t-1}$$

where we have first used $1+x \leq e^x$ and then $a^x \leq 1+a$ for any $x \in [0,1]$ with $x = \frac{s_t}{s_{t-1}} \leq 1$. By law of iterated expectation, we obtain

$$E[V_t] = E[E[V_t|\mathcal{F}_{t-1}]] \leq E[1+V_{t-1}] = 1+E[V_{t-1}].$$

Meanwhile, note that

$$E[V_1] = \exp\left(\frac{s_1}{K}\left(z_i(1)-\hat{z}_i(1)-\frac{\alpha}{2\left(q_i(1)+\frac{\gamma}{K}\right)\sqrt{NT}}\right)\right)$$

$$= \exp\left(\frac{s_1}{K}\left(z_i(1)-\hat{z}_i(1)-\frac{\alpha}{2\left(\frac{1}{N}+\frac{\gamma}{K}\right)\sqrt{NT}}\right)\right)$$

$$\leq \exp\left(\frac{s_1}{K}\left(-\frac{\alpha}{2\left(\frac{1}{N}+\frac{\gamma}{K}\right)\sqrt{NT}}\right)\right) < 1$$

where the first inequality holds by using the fact that

$$z_i(1)-\hat{z}_i(1) = \sum_{j=1}^{K}\xi_i^j(1)x_j(1)(1-\frac{1}{(1-\gamma)\frac{1}{N}\sum_{i'=1}^{N}\xi_{i'}^j(1)+\frac{\gamma}{K}})$$

$$\leq \sum_{j=1}^{K}\xi_i^j(1)(1-\frac{1}{(1-\gamma)+\frac{\gamma}{K}})$$

$$= \frac{(-1+\frac{1}{K})\gamma}{1-\gamma+\frac{\gamma}{K}} < 0$$

since $0 < x_j(1) \leq 1$ and $0 \leq \xi_{i'}^j(1) \leq 1$ and the second inequality is a result of $\alpha > 0$, $s_1 > 0$.

Therefore, by induction we have that $E[V_T] \leq T$.

To conclude, combining all above, we have that $P\left(\hat{G}_i + \alpha\hat{\sigma}_i < G_i\right) \leq e^{-\frac{\alpha^2}{4K}}E[V_T] \leq e^{-\frac{\alpha^2}{4K}}T$ and the lemma follows as we choose specific $\alpha$ that satisfies $e^{-\frac{\alpha^2}{4K}}T \leq \frac{\delta}{N}$, i.e $2\sqrt{K\ln\frac{NT}{\delta}} \leq \alpha$. $\qquad\square$

## D.2 Proof of Lemma 2

*Proof.* For simplicity, let $\vartheta = \frac{\gamma}{3K}$ and consider any sequence $i_1, \ldots, i_T$ of actions by EXP4.P. Since $p_j(t) > \frac{\gamma}{K}$, we observe

$$\hat{z}_i(t) = \sum_{j=1}^{K} \xi_j^i(t) \hat{x}_j(t) \leq \sum_{j=1}^{K} \xi_j^i(t) \frac{1}{p_j(t)} \leq \sum_{j=1}^{K} \xi_j^i(t) \frac{K}{\gamma} = \frac{K}{\gamma}.$$

Then the term $\vartheta \cdot \left( \hat{z}_i(t) + \frac{\alpha}{\frac{w_i(t)}{\sum_{j=1}^{N} w_j(t)} \sqrt{NT}} \right)$ is less than 1, noting that

$$\vartheta \cdot \left( \hat{z}_i(t) + \frac{\alpha}{\left( \frac{w_i(t)}{\sum_{j=1}^{N} w_j(t)} + \frac{\gamma}{K} \right) \sqrt{NT}} \right)$$

$$= \frac{\gamma}{3K} \left( \hat{z}_i(t) + \frac{\alpha}{\left( q_i(t) + \frac{\gamma}{K} \right) \sqrt{NT}} \right)$$

$$\leq \frac{\gamma}{3K} \cdot \frac{K}{\gamma} + \frac{\gamma}{3K} \cdot \frac{K}{\gamma} \cdot \frac{\alpha}{\sqrt{NT}}$$

$$\leq \frac{1}{3} + \frac{1}{3} \cdot \frac{2\sqrt{NT}}{\sqrt{NT}} = 1.$$

We denote $W_t = \sum_{i=1}^{N} w_i(t)$, which satisfies

$$\frac{W_{t+1}}{W_t} = \sum_{i=1}^{N} \frac{w_i(t+1)}{W_t}$$

$$= \sum_{i=1}^{N} q_i(t) \cdot \exp \left( \vartheta \cdot \left( \hat{z}_i(t) + \frac{\alpha}{\left( \frac{w_i(t)}{\sum_{j=1}^{N} w_j(t)} + \frac{\gamma}{K} \right) \sqrt{NT}} \right) \right)$$

$$\leq \sum_{i=1}^{N} q_i(t) \cdot \left( 1 + \vartheta \hat{z}_i(t) + \frac{\alpha \vartheta}{\left( q_i(t) + \frac{\gamma}{K} \right) \sqrt{NT}} + \right.$$

$$\left. 2\vartheta^2 \left( \hat{z}_i(t) \right)^2 + 2 \frac{\alpha^2 \vartheta^2}{\left( q_i(t) + \frac{\gamma}{K} \right)^2 NT} \right)$$

$$= 1 + \vartheta \sum_{i=1}^{N} q_i(t) \hat{z}_i(t) + 2\vartheta^2 \sum_{i=1}^{N} q_i(t) \left( \hat{z}_i(t) \right)^2 +$$

$$\alpha \vartheta \sum_{i=1}^{N} \frac{q_i(t)}{\left( q_i(t) + \frac{\gamma}{K} \right) \sqrt{NT}} + 2\alpha^2 \vartheta^2 \sum_{i=1}^{N} \frac{q_i(t)}{\left( q_i(t) + \frac{\gamma}{K} \right)^2 NT} \tag{3}$$

where the last inequality using the facts that $e^x < 1 + x + x^2$ for $x < 1$ and $2(a^2 + b^2) > (a+b)^2$. Note that the second term in the above expression satisfies

$$\sum_{i=1}^{N} q_i(t) \hat{z}_i(t) = \sum_{i=1}^{N} q_i(t) \left( \sum_{j=1}^{K} \xi_i^j(t) \hat{x}_j(t) \right)$$

$$= \sum_{j=1}^{K} \left( \sum_{i=1}^{N} q_i(t) \xi_i^j(t) \right) \hat{x}_j(t)$$

$$= \sum_{j=1}^{K} \left( \frac{p_j(t) - \frac{\gamma}{K}}{1 - \gamma} \right) \hat{x}_j(t) \leq \frac{x_{i_t}(t)}{1 - \gamma}$$

Also the third term yields

$$\sum_{i=1}^{N} q_i(t) \left(\hat{z}_i(t)\right)^2 = \sum_{i=1}^{N} q_i(t) \left(\sum_{j=1}^{K} \xi_i^j(t) \hat{x}_j(t)\right)^2$$

$$= \sum_{i=1}^{N} q_i(t) \left(\xi_i^{i_t} \hat{x}_{i_t}(t)\right)^2$$

$$\leq \hat{x}_{i_t}(t)^2 \frac{p_{i_t}(t)}{1-\gamma} \leq \frac{\hat{x}_{i_t}(t)}{1-\gamma}.$$

We also note that

$$\alpha\vartheta \sum_{i=1}^{N} \frac{q_i(t)}{\left(q_i(t) + \frac{\gamma}{K}\right)\sqrt{NT}} \leq \alpha\vartheta\sqrt{\frac{N}{T}}$$

and

$$2\alpha^2\vartheta^2 \sum_{i=1}^{N} \frac{q_i(t)}{\left(q_i(t) + \frac{\gamma}{K}\right)^2 NT} \leq 2\alpha^2\vartheta^2 \frac{1}{3\vartheta T} = \frac{2\alpha^2\vartheta}{3T}.$$

Plugging these estimates in (3), we get

$$\frac{W_{t+1}}{W_t} \leq 1 + \frac{\vartheta}{1-\gamma} x_{i_t}(t) + \frac{2\vartheta^2}{1-\gamma} \sum_{j=1}^{K} \hat{x}_j(t) + \alpha\vartheta\sqrt{\frac{N}{T}} + \frac{2\alpha^2\vartheta}{3T}.$$

Then we note that for any $j$, $\sum_{i=1}^{N} \xi_i^j(t) \geq \frac{1}{K}$ by the assumption that a uniform expert is included, which gives us that

$$\frac{W_{t+1}}{W_t} \leq 1 + \frac{\vartheta}{1-\gamma} x_{i_t}(t) + \frac{2\vartheta^2}{1-\gamma} K \sum_{i=1}^{N} \hat{z}_i(t) + \alpha\vartheta\sqrt{\frac{N}{T}} + \frac{2\alpha^2\vartheta}{3T}.$$

Since $\ln(1+x) < x$, we have that

$$\ln\frac{W_{t+1}}{W_t} \leq \frac{\vartheta}{1-\gamma} x_{i_t}(t) + \frac{2\vartheta^2}{1-\gamma} K \sum_{i=1}^{N} \hat{z}_i(t) + \alpha\vartheta\sqrt{\frac{N}{T}} + \frac{2\alpha^2\vartheta}{3T}.$$

Then summing over $t$ leads to

$$\ln\frac{W_{T+1}}{W_1} \leq \frac{\vartheta}{1-\gamma} G_{EXP4.P} + \frac{2\vartheta^2}{1-\gamma} K \sum_{i=1}^{N} \hat{G}_i(t) + \alpha\vartheta\sqrt{NT} + \frac{2\alpha^2\vartheta}{3}.$$

Meanwhile, by initialization we have that

$$\ln(W_1) = \ln(Nw_i(1))$$

$$= \ln\left(N \cdot \exp\left(\alpha\vartheta\sqrt{NT}\right)\right)$$

$$= \ln N + \alpha\vartheta\sqrt{NT}.$$

For any $\bar{i}$, we also have

$$
\begin{aligned}
\ln W_{T+1} &= \ln \sum_{i=1}^{N} w_i(T+1) \\
&\geq \ln w_{\bar{i}}(T+1) \\
&= \ln \left( w_{\bar{i}}(T) \exp \left( \vartheta \left( \hat{z}_{\bar{i}}(T) + \frac{\alpha}{(q_{\bar{i}}(T) + \frac{\gamma}{K})\sqrt{NT}} \right) \right) \right) \\
&= \ln \left( w_{\bar{i}}(1) \prod_{t=1}^{T} \exp \left( \vartheta \left( \hat{z}_{\bar{i}}(t) + \frac{\alpha}{(q_{\bar{i}}(t) + \frac{\gamma}{K})\sqrt{NT}} \right) \right) \right) \\
&= \ln \left( w_{\bar{i}}(1) \exp \left( \vartheta \left( \sum_{t=1}^{T} \hat{z}_{\bar{i}}(t) + \sum_{t=1}^{T} \frac{\alpha}{(q_{\bar{i}}(t) + \frac{\gamma}{K})\sqrt{NT}} \right) \right) \right) \\
&= \ln \left( \exp\left(\vartheta\alpha\sqrt{NT}\right) \exp \left( \vartheta \left( \sum_{t=1}^{T} \hat{z}_{\bar{i}}(t) + \sum_{t=1}^{T} \frac{\alpha}{(q_{\bar{i}}(t) + \frac{\gamma}{K})\sqrt{NT}} \right) \right) \right) \\
&= \ln \left( \exp \left( \vartheta \left( \sum_{t=1}^{T} \hat{z}_{\bar{i}}(t) + \alpha \left( \sqrt{NT} + \sum_{t=1}^{T} \frac{1}{(q_{\bar{i}}(t) + \frac{\gamma}{K})\sqrt{NT}} \right) \right) \right) \right) \\
&= \vartheta \hat{G}_{\bar{i}} + \alpha \vartheta \hat{\sigma}_{\bar{i}}(T+1).
\end{aligned}
$$

Therefore, we have that

$$
\vartheta \hat{G}_{\bar{i}} + \alpha \vartheta \hat{\sigma}_{\bar{i}}(T+1) - \ln N - \alpha \vartheta \sqrt{NT} \leq \frac{\vartheta}{1-\gamma} G_{EXP4.P} + \frac{2\vartheta^2}{1-\gamma} K \sum_{i=1}^{N} \hat{G}_i + \alpha \vartheta \sqrt{NT} + \frac{2\alpha^2 \vartheta}{3}.
$$

By re-organizing the terms and then multiplying by $\frac{1-\gamma}{\vartheta}$ on both sides, the above expression can be written as

$$
\begin{aligned}
G_{EXP4.P} &= (1-\gamma)(\hat{G}_{\bar{i}} + \alpha \hat{\sigma}_{\bar{i}}(T+1)) \\
&\quad - (1-\gamma)\frac{\ln N}{\vartheta} - (1-\gamma)2\alpha\sqrt{NT} - 2\vartheta K \sum_{i=1}^{N} \hat{G}_i - (1-\gamma)\frac{2\alpha^2}{3} \\
&\geq (1-\gamma)(\hat{G}_{\bar{i}} + \alpha \hat{\sigma}_{\bar{i}}(T+1)) - \frac{\ln N}{\vartheta} - 2\alpha\sqrt{NT} - 2\vartheta K \sum_{i=1}^{N} \hat{G}_i - 2\alpha^2
\end{aligned}
$$

Note that the above holds for any $\bar{i}$ and that $\sum_{i=1}^{N} \hat{G}_i \leq NU$.

The lemma follows by replacing $\hat{G}_{\bar{i}} + \alpha \hat{\sigma}_{\bar{i}}(T+1)$ with $U$ by selecting $\bar{i}$ to be the expert where $U$ achieves maximum and $\sum_{i=1}^{N} \hat{G}_i$ with $NU$.

$\square$

### D.3 PROOF OF THEOREM 1

*Proof.* Without loss of generality, we assume $\delta \geq NTe^{-\frac{NT}{K}}$ and $T \geq \max\left(\frac{3(2N+3)K\ln N}{3}, \frac{36K\ln N}{2N+3}\right)$. If either of the conditions does not hold, it is easy to observe that the theorem holds as follows. Since reward is between 0 and 1, the regret is always less or equal to $T$. On the other hand, if one of these conditions is not met, a straightforward derivation shows that the last term in the upper bound of the regret statement in the theorem is greater or equal to $T$.

By Lemma 2, we have that

$$
G_{EXP4.P} \geq \left( 1 - (1 + \frac{2N}{3})\gamma \right) \cdot U - \frac{3K}{\gamma} \ln N - 2\alpha K \sqrt{NT} - 2\alpha^2.
$$

Since $\delta \geq NTe^{-\frac{NT}{K}}$, we have $2\sqrt{K \ln\left(\frac{NT}{\delta}\right)} \leq 2\sqrt{NT}$. Then Lemma 1 gives us that

$$U \geq G_{max} = \max_i G_i, \text{ with probability at least } 1 - \delta$$

when $\gamma < \frac{1}{2}$.

Combining the two together and using the fact that $G_{max} \leq T$, we get

$$G_{max} - G_{EXP4.P} \leq \left(\left(\frac{2N}{3} + 1\right)\gamma\right) G_{max} + \frac{3K \ln N}{\gamma} + 2\alpha K\sqrt{NT} + 2\alpha^2$$

$$\leq \left(\left(\frac{2N}{3} + 1\right)\gamma\right) T + \frac{3K \ln N}{\gamma} + 2\alpha K\sqrt{NT} + 2\alpha^2, \qquad (4)$$

which holds with probability at least $1 - \delta$ when $1 - \frac{2N+3}{3} \cdot \gamma \geq 0$

Let $\gamma = \sqrt{\frac{3K \ln N}{T\left(\frac{2N}{3}+1\right)}}$ and $\alpha = 2\sqrt{K \ln\left(\frac{NT}{\delta}\right)}$. Note that $T \geq \max\left(\frac{3(2N+3)K \ln N}{3}, \frac{36K \ln N}{2N+3}\right)$, which implies that

$$1 - \frac{2N+3}{3} \cdot \gamma = 1 - \frac{2N+3}{3} \cdot \sqrt{\frac{3K \ln N}{T\left(\frac{2N}{3}+1\right)}} \geq 0$$

$$\gamma = \sqrt{\frac{3K \ln N}{T\left(\frac{2N}{3}+1\right)}} < \frac{1}{2}$$

By plugging them into the right hand side of (4), we get

$$G_{max} - G_{EXP4.P}$$

$$\leq 2\sqrt{3KT\left(\frac{2N}{3}+1\right)\ln N} + 4K\sqrt{KNT \ln\left(\frac{NT}{\delta}\right)} + 8K \ln\left(\frac{NT}{\delta}\right)$$

$$\leq 2\sqrt{3KT\left(\frac{2N}{3}+1\right)\ln N} + 4K\sqrt{KNT \ln\left(\frac{NT}{\delta}\right)} + 8NK \ln\left(\frac{NT}{\delta}\right) \quad w.p. \text{ at least } 1 - \delta$$

i.e. $R_T = G_{max} - G_{EXP4.P} \leq O^*(\sqrt{T})$, with probability at least $1 - \delta$.

$\square$

**Lemma 3.** *Let us suppose that random variables $X_1, X_2, X_3, \ldots, X_n$ are sub-Gaussian distributed with variance proxies that are upper bounded by $\sigma_X$, but are not necessarily independent. Then we have that*

$$E[\max_{1 \leq i \leq n} |X_i|] \leq \sigma_X \sqrt{2 \log 2n}.$$

*Proof.* Consider variables $X_{-1} = -X_1, X_{-2} = -X_2, \ldots, X_{-n} = -X_n$. It is straightforward to see that they are sub-Gaussian distributed with the same variance proxy as $X_1, X_2, \ldots, X_n$, respectively.

Then we have that for any $\lambda > 0$

$$
\begin{aligned}
E[\max_{1 \leq i \leq n} |X_i|] &= E[\max_{-n \leq i \leq n} X_i] \\
&= \frac{1}{\lambda} E[\log e^{\max_{-n \leq i \leq n} X_i}] \\
&\leq \frac{1}{\lambda} \log E[e^{\max_{-n \leq i \leq n} X_i}] \\
&= \frac{1}{\lambda} \log E[\max_{-n \leq i \leq n} e^{X_i}] \\
&\leq \frac{1}{\lambda} \log E[\sum_{-n \leq i \leq n} e^{X_i}] \\
&\leq \frac{1}{\lambda} \log \sum_{-n \leq i \leq n} e^{\frac{\sigma_X^2 \lambda^2}{2}} = \frac{1}{\lambda} \log 2n e^{\frac{\sigma_X^2 \lambda^2}{2}}
\end{aligned}
\tag{5}
$$

where the first inequality holds by the Jensen's inequality, the second inequality holds by the non-negativity of $e^{X_i}$, and the third inequality uses the definition of sub-Gaussian random variables.

Choosing $\lambda = \frac{2 \log 2n}{\sigma_X^2}$ in (5) leads to $E[\max_{1 \leq i \leq n} |X_i|] \leq \sigma_X \sqrt{2 \log 2n}$, which completes the proof. $\qquad \square$

### D.4 PROOF OF THEOREM 3

*Proof.* We first consider the expected deviation of $R_T$ compared to the pseudo regret $R_T'$. Following the definition, we obtain

$$
E[|R_T - R_T'|]
$$

$$
= E[|\max_i \sum_{t=1}^{T} \sum_{j=1}^{K} \epsilon_i^j(t) y_{i,t} - \sum_{t=1}^{T} y_{a_t,t} - \max_i \sum_{t=1}^{T} \sum_{j=1}^{K} \epsilon_i^j(t) c_t^T \theta_j + \sum_{t=1}^{T} c_t^T \theta_{a_t}|]
$$

$$
= E[|\max_i \sum_{t=1}^{T} \sum_{j=1}^{K} \epsilon_i^j(t) (c_t^T \theta_j + \delta_{j,t}) - \sum_{t=1}^{T} (c_t^T \theta_{a_t} + \delta_{a_t,t}) - \max i \sum_{t=1}^{T} \sum_{j=1}^{K} \epsilon_i^j(t) c_t^T \theta_j + \sum_{t=1}^{T} c_t^T \theta_{a_t}|]
$$

$$
= E[|\max_i \sum_{t=1}^{T} \sum_{j=1}^{K} \epsilon_i^j(t) \delta_{j,t} - \sum_{t=1}^{T} \delta_{a_t,t}|]
$$

Using the triangle inequality, we derive that

$$
\begin{aligned}
A &= E[|\max_i \sum_{t=1}^{T} \sum_{j=1}^{K} \epsilon_i^j(t) \delta_{j,t} - \sum_{t=1}^{T} \delta_{a_t,t}|] \\
&\leq E[|\max_i \sum_{t=1}^{T} \sum_{j=1}^{K} \epsilon_i^j(t) \delta_{j,t}|] + E[|\sum_{t=1}^{T} \delta_{a_t,t}|] \\
&\leq E[\max_i |\sum_{t=1}^{T} \sum_{j=1}^{K} \epsilon_i^j(t) \delta_{j,t}|] + E[|\sum_{t=1}^{T} \delta_{a_t,t}|] \\
&\leq \sum_{i=1}^{N} E[|\sum_{t=1}^{T} \sum_{j=1}^{K} \epsilon_i^j(t) \delta_{j,t}|] + E[|\sum_{t=1}^{T} \delta_{a_t,t}|].
\end{aligned}
\tag{6}
$$

We observe that $\sum_{t=1}^{T} \sum_{j=1}^{K} \epsilon_i^j(t)\delta_{j,t}$ and $\sum_{t=1}^{T} \delta_{a_t,t}$ are sub-Gaussian distributed based on Lemma 5 in (32). Moreover, the variance proxy of $\sum_{t=1}^{T} \sum_{j=1}^{K} \epsilon_i^j(t)\delta_{j,t}$, $\sigma_1$, meets

$$\sigma_1^2 = \sum_{t=1}^{T} \sum_{j=1}^{K} (\epsilon_i^j(t))^2 \sigma_{j,t}^2$$

$$\leq \sum_{t=1}^{T} \sum_{j=1}^{K} (\epsilon_i^j(t))^2 \sigma^2 \leq \sigma^2 \sum_{t=1}^{T} \cdot 1 = T\sigma^2$$

where the last inequality holds by the Cauchy-Schwarz inequality and the fact that $\sum_{j=1}^{K} \epsilon_i^j(t) = 1$.

Likewise, the variance proxy of $\sum_{t=1}^{T} \delta_{a_t,t}$, $\sigma_2$, meets

$$\sigma_2^2 = \sum_{t=1}^{T} \sigma_{a_t,t}^2 \leq \sum_{t=1}^{T} \sigma^2 = T\sigma^2.$$

By Lemma 3, we obtain that

$$E[|\sum_{t=1}^{T} \sum_{j=1}^{K} \epsilon_i^j(t)\delta_{j,t}|] \leq \sqrt{T\sigma^2}\sqrt{2\log 2}$$

and

$$E[|\sum_{t=1}^{T} \delta_{a_t,t}|] \leq \sqrt{T\sigma^2}\sqrt{2\log 2}.$$

Subsequently, we derive that

$$A \leq N(\sqrt{T\sigma^2}\sqrt{2\log 2}) + \sqrt{T\sigma^2}\sqrt{2\log 2} = (N+1)\sigma\sqrt{2T\log 2}$$

which immediately implies that

$$E[|R_T - R_T'|] \leq (N+1)\sigma\sqrt{2T\log 2} = O^*(\sqrt{T}). \tag{7}$$

We next decompose the expected regret $E[R_T]$ as follows. Note that

$$E[R_T] = E[R_T 1_{R_T \geq O^*(\sqrt{T})} + R_T 1_{R_T \leq O^*(\sqrt{t})}]$$

$$\leq E[R_T 1_{R_T \geq O^*(\sqrt{T})}] + O^*(\sqrt{T})P(R_T \leq O^*(\sqrt{T}))$$

$$\leq E[R_T 1_{R_T \geq O^*(\sqrt{T}) + E[R_T]}] + E[R_T 1_{O^*(\sqrt{T}) \leq R_T \leq O^*(\sqrt{T}) + E[R_T]}] + O^*(\sqrt{T})$$

$$:= E_1 + E_2 + O^*(\sqrt{T}). \tag{8}$$

Let $P_1 = P\left(R_T \leq \log(1/\delta)O^*(\sqrt{T})\right)$ which equals to $P\left(R_T \leq O^*(\sqrt{T})\right)$ since $\log(1/\delta) = \log(\sqrt{T}) = O^*(\sqrt{T})$. By Theorem 2 we have

$$P_1 = (1-\delta) \cdot (1-\eta)^T. \tag{9}$$

We consider $\delta = 1/\sqrt{T}$ and $\eta = T^{-a}$ for $a > 2$. We have

$$\lim_{T\to\infty} (1-\delta)(1-\eta)^T = \lim_{T\to\infty} (1-\delta)(1-\frac{1}{T^a})^T$$

$$= \lim_{T\to\infty} (1-\delta)(1-\frac{1}{T^a})^{(T^a)\cdot\frac{T}{T^a}} = \lim_{T\to\infty} e^{\frac{T}{T^a}}$$

and

$$\lim_{T\to\infty} \left(1 - (1-\delta)(1-\eta)^T\right) \cdot \log T \cdot T = \lim_{T\to\infty} (1 - e^{\frac{T}{T^a}}) \cdot \log(T) \cdot T$$

$$\leq \lim_{T\to\infty} \log(T) \cdot T \cdot T^{1-a} = \lim_{T\to\infty} T^{2-a} \cdot \log(T) = 0. \tag{10}$$

By using (7), (9), and (10), we obtain

$$
\begin{aligned}
E_1 &= E\left[R_T \mathbb{1}_{R_T \geq O^*(\sqrt{T}) + E[R_T]}\right] \\
&= E\left[(R_T - R_T')\,\mathbb{1}_{(R_T - E[R_T]) \geq O^*(\sqrt{T})}\right] + E\left[R_T'\mathbb{1}_{(R_T - E[R_T]) \geq O^*(\sqrt{T})}\right] \\
&\leq E\left[|R_T - R_T'|\right] + R_T' \cdot P\left(R_T \geq E[R_T] + O^*(\sqrt{T})\right) \\
&\leq E\left[|R_T - R_T'|\right] + E[R_T] \cdot P\left(R_T \geq E[R_T] + O^*(\sqrt{T})\right) \\
&\leq O^*(\sqrt{T}) + C_0 \cdot \log(T) \cdot T \cdot P\left(R_T \geq O^*(\sqrt{T})\right) \\
&= O^*(\sqrt{T}) + C_0 \cdot \log(T) \cdot T\,(1 - P_1) = O^*(\sqrt{T}) \tag{11}
\end{aligned}
$$

where the second inequality uses the Jensen's inequality which gives us

$$
R_T' \leq E[R_T].
$$

Additionally, we note that by definition,

$$
\begin{aligned}
E[R_T] &= E[\max_i \sum_{t=1}^{T} \sum_{j=1}^{K} \epsilon_i^j(t) y_{i,t} - \sum_{t=1}^{T} y_{a_t,t}] \\
&\leq E[|\max_i \sum_{t=1}^{T} \sum_{j=1}^{K} \epsilon_i^j(t) y_{i,t}|] + E[\sum_{t=1}^{T} y_{a_t,t}] \\
&\leq T \cdot N \cdot K E[\max_{j,t} y_{i,t}] + T E[\max_{j,t} y_{i,t}] \\
&= T(NK + 1) E[\max_{j,t} y_{i,t}] \\
&\leq T(NK + 1)(\max_{j,t} c_t^T \theta_j + E[\max_{j,t} \delta_{i,t}]) \\
&\leq T(NK + 1)(1 + \sigma\sqrt{2\log{(2T)K}}) \leq C_L \cdot T \cdot \log T \tag{12}
\end{aligned}
$$

where the last inequality holds by the fact that $||c_t|| \leq 1, ||\theta_j|| \leq 1$ and by Lemma 3. Here $C_L$ is a constant.

Consequently, the asymptotic behavior of the second term $E_2$ reads

$$
\begin{aligned}
E_2 &= E\left[R_T \mathbb{1}_{O^*(\sqrt{T}) < R_T < O^*(\sqrt{T}) + E[R_T]}\right] \\
&= E\left[R_T \mathbb{1}_{R_T - O^*(\sqrt{T}) \in (0, E[R_T])}\right] \\
&= E\left[\left(R_T - O^*(\sqrt{T})\right)\mathbb{1}_{R_T - O^*(\sqrt{T}) \in (0, E[R_T])}\right] + O^*(\sqrt{T}) \\
&\leq E[R_T] P\left(R_T - O^*(\sqrt{T}) \in (0, E[R_T])\right) + O^*(\sqrt{T}) \\
&\leq E[R_T] P\left(R_T - O^*(\sqrt{T}) > 0\right) + O^*(\sqrt{T}) \\
&\leq C_L \log(T) \cdot T \cdot (1 - P_1) + O^*(\sqrt{T}) = O^*(\sqrt{T}) \tag{13}
\end{aligned}
$$

where the last inequality uses (9) and (12).

Combining all these together, we obtain

$$
E[R_T] \leq O^*(\sqrt{T}) + O^*(\sqrt{T}) + O^*(\sqrt{T}) = O^*(\sqrt{T})
$$

which concludes the proof.

$\square$

### D.5 PROOF OF THEOREM 4

*Proof.* By the definition of $R_T^{simple}$, we have that

$$R_T = \max_i \sum_{t=1}^{T} \sum_{j=1}^{K} \epsilon_j^i(t) y_{i,t} - \sum_{t=1}^{T} y_{a_t,t}$$

$$= \max_i \sum_{t=1}^{T} \sum_{j=1}^{K} \epsilon_j^i(t)(c_t^T \theta_j + \delta_{j,t}) - \sum_{t=1}^{T} (c_t^T \theta_{a_t} + \delta_{a_t,t})$$

$$\leq \max_i \sum_{t=1}^{T} \sum_{j=1}^{K} \epsilon_j^i(t) c_t^T \theta_j + \max_i \sum_{t=1}^{T} \sum_{j=1}^{K} \epsilon_j^i(t) \delta_{j,t} - \sum_{t=1}^{T} c_t^T \theta_{a_t} - \sum_{t=1}^{T} \delta_{a_t,t}$$

$$\leq \sum_{t=1}^{T} \max_i \sum_{j=1}^{K} \epsilon_i^j(t) c_t^T \theta_j + \max_i \sum_{t=1}^{T} \sum_{j=1}^{K} \epsilon_j^i(t) \delta_{j,t} - \sum_{t=1}^{T} c_t^T \theta_{a_t} - \sum_{t=1}^{T} \delta_{a_t,t}$$

$$\leq \sum_{t=1}^{T} \max_i \sum_{j=1}^{K} \epsilon_i^j(t) \max_j c_t^T \theta_j + \max_i \sum_{t=1}^{T} \sum_{j=1}^{K} \epsilon_j^i(t) \delta_{j,t} - \sum_{t=1}^{T} c_t^T \theta_{a_t} - \sum_{t=1}^{T} \delta_{a_t,t}$$

$$= \sum_{t=1}^{T} (\max_j c_t^T \theta_j) \max_i \sum_{j=1}^{K} \epsilon_i^j(t) + \max_i \sum_{t=1}^{T} \sum_{j=1}^{K} \epsilon_j^i(t) \delta_{j,t} - \sum_{t=1}^{T} c_t^T \theta_{a_t} - \sum_{t=1}^{T} \delta_{a_t,t}$$

$$= \sum_{t=1}^{T} \max_j c_t^T \theta_j + \max_i \sum_{t=1}^{T} \sum_{j=1}^{K} \epsilon_j^i(t) \delta_{j,t} - \sum_{t=1}^{T} c_t^T \theta_{a_t} - \sum_{t=1}^{T} \delta_{a_t,t}$$

$$= R_T^{cum} + \max_i \sum_{t=1}^{T} \sum_{j=1}^{K} \epsilon_j^i(t) \delta_{j,t} - \sum_{t=1}^{T} \delta_{a_t,t}$$

where the second and third inequalities hold by the Jensen's inequality, and the last inequality uses the definition of $R_T^{cum}$ which is defined by

$$R_T^{cum} = \sum_{t=1}^{T} \max_j c_t^T \theta_j - \sum_{t=1}^{T} c_t^T \theta_{a_t}.$$

Subsequently, we obtain that

$$E[R_T] \leq E[R_T^{cum} + \max_i \sum_{t=1}^{T} \sum_{j=1}^{K} \epsilon_j^i \delta_{j,t} - \sum_{t=1}^{T} \delta_{a_t,t}]$$

$$= E[R_T^{cum}] + E[\max_i \sum_{t=1}^{T} \sum_{j=1}^{K} \epsilon_j^i \delta_{j,t}]$$

$$\leq E[R_T^{cum}] + \sqrt{\log N \sigma(\sum_{t=1}^{T} \sum_{j=1}^{K} \epsilon_j^i \delta_{j,t})}$$

$$\leq E[R_T^{cum}] + \sqrt{\log N} \sqrt{TK}$$

where the first equality holds by the fact that $\delta_{a_t,t}$ has mean 0, the second inequality uses Lemma 3, and the last inequality results from Lemma 5 in (32).

This implies that if $E[R_T^{cum}]$ is upper bounded by $G(T)$, then we have $E[R_T] \leq \max\{O^*(\sqrt{T}), G(T)\}$, which completes the proof of the first half of the statement.

On the other hand, again based on the definition of $E[R_T]$, we have

$$R_T = \max_i \sum_{t=1}^{T} \sum_{j=1}^{K} \epsilon_j^i(t)(c_t^T \theta_j + +\delta_{j,t}) - \sum_{t=1}^{T}(c_t^T \theta_{a_t} + \delta_{a_t,t})$$

$$\geq \sum_{t=1}^{T} \sum_{j=1}^{K} \pi_j(c_t^T \theta_j + \delta_{j,t}) - \sum_{t=1}^{T}(c_t^T \theta_{a_t} + \delta_{a_t,t}),$$

which leads to

$$E[R_T] \geq E[\sum_{t=1}^{T} \sum_{j=1}^{K} \pi_j c_t^T \theta_j] + E[\sum_{t=1}^{T} \sum_{j=1}^{K} \pi_j \delta_{j,t}] - E[\sum_{t=1}^{T} c_t^T \theta_{a_t}] - E[\delta_{a_t,t}]$$

$$= E[\sum_{t=1}^{T} \sum_{j=1}^{K} \pi_j c_t^T \theta_j] - E[\sum_{t=1}^{T} c_t^T \theta_{a_t}] + E[\sum_{t=1}^{T} \sum_{j=1}^{K} \pi_j \delta_{j,t}]$$

$$= E[\sum_{t=1}^{T} \sum_{j=1}^{K} \pi_j c_t^T \theta_j] - E[\sum_{t=1}^{T} c_t^T \theta_{a_t}]$$

where the last equality uses the fact that $\delta_{j,t}$ is independent of everything else, including $\pi_j$.

By assumption, we obtain

$$\sum_{t=1}^{T} \sum_{j=1}^{K} \pi_j^t c_t^T \theta_j \geq \sum_{t=1}^{T} \max_j \mu_{j,t} - F(T) = \sum_{t=1}^{T} \max_j c_t^T \theta_j - F(T)$$

which immediately implies that

$$E[R_T] \geq E[\sum_{t=1}^{T} \max_j c_t^T \theta_j] - F(T) - E[\sum_{t=1}^{T} c_t^T \theta_{a_t}]$$

$$= E[R_T^{cum}] - F(T).$$

Henceforth, if the simple regret satisfies that $E[R_T] \leq O^*(\sqrt{T})$, which holds by Theorem 3, then the cumulative regret also meets $E[R_T^{cum}] \leq \max\{O^*(\sqrt{T}), F(T)\}$.

This completes the proof of the second half of the statement.

$\square$

### D.6 PROOF OF THEOREM 5

*Proof.* Since the rewards can be unbounded in our setting, we consider truncating the reward with any $\Delta > 0$ for any arm $i$ by $r_i^t = \bar{r}_i^t + \hat{r}_i^t$ where

$$\bar{r}_i^t = r_i^t \cdot \mathbb{1}_{(-\Delta \leq r_i^t \leq \Delta)}, \hat{r}_i^t = r_i^t \cdot \mathbb{1}_{(|r_i^t| > \Delta)}.$$

Then for any parameter $0 < \eta < 1$, we choose such $\Delta$ that satisfies

$$P(r_i^t = \bar{r}_i^t, i \leq K) = P(-\Delta \leq r_1^t \leq \Delta, \ldots, -\Delta \leq r_K^t \leq \Delta)$$

$$= \int_{-\Delta}^{\Delta} \int_{-\Delta}^{\Delta} \ldots \int_{-\Delta}^{\Delta} f(x_1, \ldots, x_K) dx_1 \ldots dx_K \geq 1 - \eta. \tag{14}$$

The existence of such $\Delta = \Delta(\eta)$ follows from elementary calculus.

Let $A = \{|r_i^t| \leq \Delta$ for every $i \leq K, t \leq T\}$. Then the probability of this event is

$$P(A) = P(r_i^t = \bar{r}_i^t, i \leq K, t \leq T) \geq (1 - \eta)^T.$$

With probability $(1 - \eta)^T$, the rewards of the player are bounded in $[-\Delta, \Delta]$ throughout the game. Then $R_T^{c,B} = \max_i \sum_{t=1}^{T} \sum_{j=1}^{K} \xi_i^j(t) \bar{r}_j^t - \sum_{t=1}^{T} \bar{r}_t^i \leq T \cdot \Delta - \sum_{t=1}^{T} r_t$ is the regret under event $A$, i.e. $R_T^c = R_T^{c,B}$ with probability $(1 - \eta)^T$. For the EXP4.P algorithm and $R_T^{c,B}$ with rewards $\bar{r}_j^t = \frac{\bar{r}_j^t + \Delta}{\Delta}$ satisfying $0 < \bar{r}_j^t < 1$, for every $\delta > 0$, according to Theorem 1 , we have

$$R_T^{c,B} \leq 4\Delta(\eta) \left( 2\sqrt{3KT\left(\frac{2N}{3} + 1\right)\ln N} + 4K\sqrt{KNT\ln\left(\frac{NT}{\delta}\right)} + 8NK\ln\left(\frac{NT}{\delta}\right) \right).$$

Then we have

$$R_T \leq 4\Delta(\eta) \left( 2\sqrt{3KT\left(\frac{2N}{3} + 1\right)\ln N} + 4K\sqrt{KNT\ln\left(\frac{NT}{\delta}\right)} + 8NK\ln\left(\frac{NT}{\delta}\right) \right)$$

with probability $(1 - \delta) \cdot (1 - \eta)^T$.

$\square$

## D.7 PROOF OF THEOREM 6

**Lemma 4.** *For any non-decreasing differentiable function $\Delta = \Delta(T) > 0$ satisfying*

$$\lim_{T \to \infty} \frac{\Delta(T)^2}{\log(T)} = \infty, \qquad \lim_{T \to \infty} \Delta'(T) \leq C_0 < \infty,$$

*and any $0 < \delta < 1, a > 2$ we have*

$$P\left( R_T^c \leq \Delta(T) \cdot \log(1/\delta) \cdot O^*(\sqrt{T}) \right) \geq (1 - \delta)\left(1 - \frac{1}{T^a}\right)^T$$

*for any $T$ large enough.*

*Proof.* Let $a > 2$ and let us denote

$$F(y) = \int_{-y}^{y} f(x_1, x_2, \ldots, x_K) dx_1 dx_2 \ldots dx_K,$$

$$\zeta(T) = F\left(\Delta(T) \cdot \mathbf{1}\right) - \left(1 - \frac{1}{T^a}\right)$$

for $y \in \mathbb{R}^K$ and $\mathbf{1} = (1, \ldots, 1) \in \mathbb{R}^K$. Let also $y_{-i} = (y_1, \ldots, y_{i-1}, y_{i+1}, \ldots, y_K)$ and $x|_{x_i=y} = (x_1, \ldots, x_{i-1}, y, x_{i+1}, \ldots, x_K)$. We have $\lim_{T \to \infty} \zeta(T) = 0$.

The gradient of $F$ can be estimated as

$$\nabla F \leq \left( \int_{-y_{-1}}^{y_{-1}} f\left(x|_{x_1=y_1}\right) dx_2 \ldots dx_K, \ldots, \int_{-y_{-K}}^{y_{-K}} f\left(x|_{x_K=y_K}\right) dx_1 \ldots dx_{K-1} \right).$$

According to the chain rule and since $\Delta'(T) \geq 0$, we have

$$\frac{dF(\Delta(T) \cdot \mathbf{1})}{dT} \leq \int_{-\Delta(T) \cdot \mathbf{1}_{-1}}^{\Delta(T) \cdot \mathbf{1}_{-1}} f\left(x|_{x_1=\Delta(T)}\right) dx_2 \ldots dx_K \cdot \Delta'(T) +$$

$$\ldots + \int_{-\Delta(T) \cdot \mathbf{1}_{-K}}^{\Delta(T) \cdot \mathbf{1}_{-K}} f\left(x|_{x_K=\Delta(T)}\right) dx_1 \ldots dx_{K-1} \cdot \Delta'(T).$$

Next we consider

$$\int_{-\Delta(T)\mathbf{1}_{-i}}^{\Delta(T)\mathbf{1}_{-i}} f\left(x|_{x_i=\Delta(T)}\right) dx_1 \ldots dx_{i-1} dx_{i+1} \ldots dx_K$$

$$\leq e^{-\frac{1}{2}a_{ii}(\Delta(T))^2 + \mu_i \Delta(T)} \cdot \int_{-\Delta(T)\mathbf{1}_{-i}}^{\Delta(T)\mathbf{1}_{-i}} e^{g(x_{-i})} dx_1 \ldots dx_{i-1} dx_{i+1} \ldots dx_K.$$

Here $e^{g(x_{-i})}$ is the conditional density function given $x_i = \Delta(T)$ and thus $\int_{-\Delta(T)\mathbf{1}_{-i}}^{\Delta(T)\mathbf{1}_{-i}} e^{g(x_{-i})} dx_1 \ldots dx_{i-1} dx_{i+1} \ldots dx_K \leq 1$. We have

$$\int_{-\Delta(T)\mathbf{1}_{-i}}^{\Delta(T)\mathbf{1}_{-i}} f\left(x|_{x_i=\Delta(T)}\right) dx_1 \ldots dx_{i-1} dx_{i+1} \ldots dx_K$$
$$\leq e^{-\frac{1}{2} a_{ii}(\Delta(T))^2 + \mu_i \Delta(T)}$$
$$\leq e^{-\frac{1}{2} \min_j a_{jj}(\Delta(T))^2 + \max_j \mu_j \Delta(T)}.$$

Then for $T \geq T_0$ we have $\Delta'_T \leq C_0 + 1$ and in turn

$$\zeta'(T) \leq (C_0 + 1) \cdot K \cdot e^{-\frac{1}{2} \min_j a_{jj}(\Delta(T))^2 + \max_j \mu_j \Delta(T)} - a \cdot T^{-a-1}.$$

Since we only consider non-degenerate sub-Gaussian bandits with $\min a_{ii} > 0$, $\mu_i$ are constants and $\Delta(T) \to \infty$ as $T \to \infty$ according to the assumptions in Lemma 4, there exits $C_1 > 0$ and $T_1$ such that

$$e^{-\frac{1}{2} \min_j a_{jj}(\Delta(T))^2 + \max_j \mu_j \Delta(T)} \leq e^{-C_1 \Delta(T)^2} \text{ for every } T > T_1.$$

Since $\lim_{T \to \infty} \frac{\Delta(T)^2}{\log(T)} = \infty$, we have

$$\Delta(T)^2 > \frac{2(a+1)}{C_1} \cdot \log(T) \text{ for } T > T_2.$$

These give us that

$$\zeta(T)' \leq (C_0 + 1)K e^{-2(a+1)\log T} - aT^{-a-1}$$
$$= (C_0 + 1)K e^{-2(a+1)\log T} - ae^{-(a+1)\log T}$$
$$< 0 \text{ for } T \geq T_3 \geq \max(T_0, T_1, T_2).$$

This concludes that $\zeta'(T) < 0$ for $T \geq T_3$. We also have $\lim_{T \to \infty} \zeta(T) = 0$ according to the assumptions. Therefore, we finally arrive at $\zeta(T) > 0$ for $T \geq T_3$. This is equivalent to

$$\int_{-\Delta(T)\cdot\mathbf{1}}^{\Delta(T)\cdot\mathbf{1}} f\left(x_1, \ldots, x_K\right) dx_1 \ldots dx_K \geq 1 - \frac{1}{T^a},$$

i.e. the rewards are bounded by $\Delta(T)$ with probability $1 - \frac{1}{T^a}$. Then by the same argument for $T$ large enough as in the proof of Theorem 1, we have

$$P\left(R_T^c \leq \Delta(T) \cdot \log(1/\delta) \cdot O^*(\sqrt{T})\right) \geq (1 - \delta)(1 - \frac{1}{T^a})^T.$$

$\square$

*Proof of Theorem 3.* In Lemma 4, we choose $\Delta(T) = \log(T)$, which meets all of the assumptions. The result now follows from $\log T \cdot O^*(\sqrt{T}) = O^*(\sqrt{T})$, Lemma 4 and Theorem 2. $\square$

### D.8 Proof of Theorem 7

We first list 3 known lemmas. The following lemma by **(author?)** (13) provides a way to bound deviations.

**Lemma 5.** *For any function class F, and i.i.d. random variable $\{x_1, x_2, \ldots, x_T\}$, the result*

$$E_x\left[\sup_{f \in F} \left|E_x f - \frac{1}{T}\sum_{t=1}^T f(x_t)\right|\right] \leq 2R_T^c(F)$$

*holds where $R_T^c(F) = E_{x,\sigma}\left[\sup_f \left|\frac{1}{T}\sum_{t=1}^T \sigma_t f(x_t)\right|\right]$ and $\sigma_t$ is a $\{-1,1\}$ random walk of $t$ steps.*

The following result holds according to **(author?)** (4).

**Lemma 6.** *For any subclass $A \subset F$, we have $\hat{R}_T^c \leq R(A, T) \cdot \frac{\sqrt{2 \log |A|}}{T}$, where $R(A, T) = \sup_{f \in A} \left( \sum_{t=1}^T f^2(x_t) \right)^{\frac{1}{2}}$ and $\hat{R}_T^c = \sup_f \left| \frac{1}{T} \sum_{t=1}^T \sigma_t f(x_t) \right|.$*

A random variable $X$ is $\sigma^2$-sub-Gaussian if for any $t > 0$, the tail probability satisfies

$$P(|X| > t) \leq B e^{-\sigma^2 t^2},$$

where $B$ is a positive constant. The following lemma is listed in the Appendix A of **(author?)** (9).

**Lemma 7.** *For i.i.d. $\sigma^2$-sub-Gaussian random variables $\{Y_1, Y_2, \ldots, Y_T\}$, we have*

$$E\left[\max_{1 \leq t \leq T} |Y_t|\right] \leq \sigma \sqrt{2 \log T} + \frac{4\sigma}{\sqrt{2 \log T}}.$$

*Proof of Theorem 4.* Let us define $F = \{f_{i,t} : x \to \sum_{j=1}^K \xi_i^j(t) x_j(t) | j = 1, 2, \ldots, K; t = 1, \ldots, T\}$. Let $x_t = x(t) = (r_1^t, r_2^t, \ldots, r_K^t)$ where $r_i^t$ is the reward of arm $i$ at step $t$ and let $a_t$ be the arm selected at time $t$ by EXP4.P. Then for any $f_{j,t} \in F$, $f_{j,t}(x_{t_i}) = I_{t=t_i} \sum_{j=1}^K \xi_i^j(t) x_j(t)$. In sub-Gaussian bandits, $\{x_1, x_2, \ldots, x_T\}$ are i.i.d. random variables since the sub-Gaussian distribution $\sigma^2 - \mathcal{N}(\mu, \Sigma)$ is invariant to time and independent of time. Then by Lemma 5, we have

$$E\left[\max_{i,t} \left| \sum_{j=1}^K \xi_i^j(t)\mu_j - \frac{1}{T} \sum_{t=1}^T \sum_{j=1}^K \xi_i^j(t) r_j^t \right|\right] \leq 2R_T^c(F).$$

We consider

$$E\left[|R_T' - R_T|\right] = E\left[\left| \sum_{t=1}^T \max_i \sum_{j=1}^K \xi_i^j(t)\mu_j - \sum_{t=1}^T \mu_{a_t} - \left( \max_i \sum_{t=1}^T \sum_{j=1}^K \xi_i^j(t) r_j^t - \sum_{t=1}^T r_{a_t}^t \right) \right|\right]$$

$$\leq E\left[\left| T \cdot \max_{i,t} \sum_{j=1}^K \xi_i^j(t)\mu_j - \max_i \sum_{t=1}^T \sum_{j=1}^K \xi_i^j(t) r_j^t - \left( \sum_{t=1}^T \mu_{a_t} - \sum_{t=1}^T r_{a_t}^t \right) \right|\right]$$

$$\leq E\left[\left| T \cdot \max_{i,t} \sum_{j=1}^K \xi_i^j(t)\mu_j - \max_i \sum_{t=1}^T \sum_{j=1}^K \xi_i^j(t) r_j^t \right|\right] + E\left[\left| \sum_{t=1}^T \mu_{a_t} - \sum_{t=1}^T r_{a_t}^t \right|\right]$$

$$\leq E\left[ T \cdot \max_{i,t} \left| \sum_{j=1}^K \xi_i^j(t)\mu_j - \frac{1}{T} \sum_{t=1}^T \sum_{j=1}^K \xi_i^j(t) r_j^t \right|\right] + E\left[\left| \sum_{t=1}^T \mu_{a_t} - \sum_{t=1}^T r_{a_t}^t \right|\right]$$

$$\leq 2T R_T^c(F) + 2T_1 R_{T_1}^c(F) + \cdots + 2T_K R_{T_K}^c(F) \tag{15}$$

where $T_i$ is the number of pulls of arm $i$. Clearly $T_1 + T_2 + \ldots + T_K = T$. By Lemma 6 with $A = F$ which has a cardinality of $NT$ we get

$$R_T^c(F) = E\left[\hat{R}_T^c(F)\right] \leq E[R(F, T)] \cdot \frac{\sqrt{2 \log (NT)}}{T},$$

$$R_{T_j}^c(F) \leq E\left[R\left(F, T_j\right)\right] \cdot \frac{\sqrt{2 \log (NT)}}{T_j} \qquad j = \{1, 2,, \ldots, K\}.$$

Since $R(F, T)$ is increasing in $T$ and $T_j \leq T$, we have $R_{T_j}^c(F) \leq E\left[R\left(F, T\right)\right] \cdot \frac{\sqrt{2 \log(NT)}}{T_j}$.

We next bound the expected deviation $E\left[|R_T' - R_T|\right]$ based on (15) as follows

$$E\left[|R_T' - R_T|\right] \leq 2T E[R(F, T)] \frac{\sqrt{2 \log (NT)}}{T} + \sum_{j=1}^K \left[ 2T_j E[R(F, T)] \frac{\sqrt{2 \log (NT)}}{T_j} \right]$$

$$\leq 2(K + 1)\sqrt{2 \log (NT)} E[R(F, T)]. \tag{16}$$

Regarding $E[R(F,T)]$, we have

$$E[R(F,T)] = E\left[\sup_{f\in F}\left(\sum_{t=1}^{T} f^2(x_t)\right)^{\frac{1}{2}}\right] = E\left[\sup_{i}\left(\sum_{t=1}^{T}\left(\sum_{j=1}^{K}\xi_i^j(t)r_j^t\right)^2\right)^{\frac{1}{2}}\right]$$

$$\leq E\left[\sup_{i,t}\left(T\cdot\left(\sum_{j=1}^{K}\xi_i^j(t)r_j^t\right)^2\right)^{\frac{1}{2}}\right] \tag{17}$$

$$\leq \sqrt{T}\cdot E\left[\sup_{i,t}\sum_{j=1}^{K}|\xi_i^j(t)r_j^t|\right]$$

$$\leq \sqrt{T}\cdot E\left[\sum_{i=1}^{N}\sup_{t}\sum_{j=1}^{K}|\xi_i^j(t)r_j^t|\right] = \sqrt{T}\cdot\sum_{i=1}^{N}E\left[\sum_{j=1}^{K}\sup_{t}|\xi_i^j(t)r_j^t|\right]$$

$$\leq \sqrt{T}\cdot\sum_{i=1}^{N}\sum_{j=1}^{K}E\left[\max_{1\leq t\leq T}|r_j^t|\right] = \sqrt{NT}\cdot\sum_{i=1}^{K}E\left[\max_{1\leq t\leq T}|r_j^t|\right]. \tag{18}$$

We next use Lemma 7 for any arm $j$. To this end let $Y_t = r_j^t$. Since $x_t$ are sub-Gaussian, the marginals $Y_t$ are also sub-Gaussian with mean $\mu_i$ and standard deviation of $a_{ii}$. Combining this with the fact that a sub-Gaussian random variable is $\sigma^2$-sub-Gaussian justifies the use of the lemma. Thus $E\left[\max_{1\leq t\leq T}|r_t^j|\right] \leq a_{j,j}\cdot\sqrt{2\log T} + \frac{4a_{j,j}}{\sqrt{2\log T}}$.

Continuing with equation 18 we further obtain

$$E[R(F,T)] \leq \sqrt{NT}\cdot K\cdot\max_{j}\left(a_{j,j}\sqrt{2\log T} + \frac{4a_{j,j}}{\sqrt{2\log T}}\right)$$

$$= \left(K\sqrt{2NT\log T} + \frac{4\sqrt{NT}}{\sqrt{2\log T}}\right)\cdot\max_{j}a_{j,j}. \tag{19}$$

By combining equation 16 and equation 19 we conclude

$$E\left[|R_T' - R_T|\right] \leq 2(K+1)\sqrt{2\log(NT)}\cdot\max_{j}a_{j,j}\cdot\left(K\sqrt{2NT\log T} + \frac{4\sqrt{NT}}{\sqrt{2\log T}}\right) \tag{20}$$

$$= O^*(\sqrt{T}).$$

We now turn our attention to the expectation of regret $E[R_T]$. It can be written as

$$E[R_T] = E\left[R_T\mathbb{1}_{R_T\leq O^*(\sqrt{T})}\right] + E\left[R_T\mathbb{1}_{R_T>O^*(\sqrt{T})}\right]$$

$$\leq O^*(\sqrt{T})P\left(R_T\leq O^*(\sqrt{T})\right) + E\left[R_T\mathbb{1}_{R_T>O^*(\sqrt{T})}\right] \leq O^*(\sqrt{T}) + E\left[R_T\mathbb{1}_{R_T>O^*(\sqrt{T})}\right]$$

$$= O^*(\sqrt{T}) + E\left[R_T\mathbb{1}_{O^*(\sqrt{T})<R_T<O^*(\sqrt{T})+E[R_T]}\right] + E\left[R_T\mathbb{1}_{R_T\geq O^*(\sqrt{T})+E[R_T]}\right]. \tag{21}$$

We consider $\delta = 1/\sqrt{T}$ and $\eta = T^{-a}$ for $a > 2$. We have

$$\lim_{T\to\infty}(1-\delta)(1-\eta)^T = \lim_{T\to\infty}(1-\delta)(1-\frac{1}{T^a})^T$$

$$= \lim_{T\to\infty}(1-\delta)(1-\frac{1}{T^a})^{(T^a)\cdot\frac{T}{T^a}} = \lim_{T\to\infty}e^{\frac{T}{T^a}}$$

and

$$
\lim_{T \to \infty} \left(1 - (1 - \delta)(1 - \eta)^T\right) \cdot \log T \cdot T = \lim_{T \to \infty} \left(1 - e^{\frac{T}{T^a}}\right) \cdot \log(T) \cdot T
$$
$$
\leq \lim_{T \to \infty} \log(T) \cdot T \cdot T^{1-a} = \lim_{T \to \infty} T^{2-a} \cdot \log(T) = 0.
$$
(22)

Let $P_1 = P\left(R_T \leq \log(1/\delta)O^*(\sqrt{T})\right)$ which equals to $P\left(R_T \leq O^*(\sqrt{T})\right)$ since $\log(1/\delta) = \log(\sqrt{T}) = O^*(\sqrt{T})$. By Theorem 3 we have $P_1 = (1 - \delta) \cdot (1 - \eta)^T$.

Note that $E[R_T] \leq C_0 \log(T) \cdot T$ as shown by

$$
E[R_T] = E\left[\max_i \sum_{t=1}^{T} \sum_{j=1}^{K} \xi_i^j(t) r_j^t - \sum_{t=1}^{T} r_{a_t}^t\right] \leq E\left[\left|\max_i \sum_{t=1}^{T} \sum_{j=1}^{K} \xi_i^j(t) r_j^t\right|\right] + E\left[\max_i \sum_{t=1}^{T} |r_i^t|\right]
$$

$$
\leq TNK \cdot E\left[\max_j \max_t |r_j^t|\right] + T \cdot E\left[\max_j \max_t |r_j^t|\right] = T(NK + 1) \cdot E\left[\max_j \max_t |r_j^t|\right]
$$

$$
\leq (NK + 1)T \cdot \sum_{j=1}^{K} E\left[\max_t |r_j^t|\right] \leq (NK + 1)T \cdot \sum_{j=1}^{K} \left(a_{j,j}\sqrt{2 \log T} + \frac{4a_{j,j}}{\sqrt{\log T}}\right)
$$

$$
\leq 2T \cdot \sum_{j=1}^{K} \max_i a_{j,j} \left(\sqrt{2 \log T} + \frac{4}{\sqrt{\log T}}\right)
$$

$$
\leq C_0 \cdot T \cdot \log(T)
$$

for a constant $C_0$.

The asymptotic behavior of the second term in equation 21 reads

$$
E\left[R_T \mathbb{1}_{O^*(\sqrt{T}) < R_T < O^*(\sqrt{T}) + E[R_T]}\right] = E\left[R_T \mathbb{1}_{R_T - O^*(\sqrt{T}) \in (0, E[R_T])}\right]
$$

$$
= E\left[\left(R_T - O^*(\sqrt{T})\right) \mathbb{1}_{R_T - O^*(\sqrt{T}) \in (0, E[R_T])}\right] + O^*(\sqrt{T})
$$

$$
\leq E[R_T] P\left(R_T - O^*(\sqrt{T}) \in (0, E[R_T])\right) + O^*(\sqrt{T})
$$

$$
\leq E[R_T] P\left(R_T - O^*(\sqrt{T}) > 0\right) + O^*(\sqrt{T})
$$

$$
\leq C_0 \log(T) \cdot T \cdot (1 - P_1) + O^*(\sqrt{T}) = O^*(\sqrt{T})
$$

where at the end we use equation 22.

Regarding the third term in equation 21, we note that $R_T' \leq E[R_T]$ by the Jensen's inequality. By using equation 20 and again equation 22 we obtain

$$
E\left[R_T \mathbb{1}_{R_T \geq O^*(\sqrt{T}) + E[R_T]}\right]
$$

$$
= E\left[(R_T - R_T') \mathbb{1}_{(R_T - E[R_T]) \geq O^*(\sqrt{T})}\right] + E\left[R_T' \mathbb{1}_{(R_T - E[R_T]) \geq O^*(\sqrt{T})}\right]
$$

$$
\leq E[|R_T - R_T'|] + R_T' \cdot P\left(R_T \geq E[R_T] + O^*(\sqrt{T})\right)
$$

$$
\leq E[|R_T - R_T'|] + E[R_T] \cdot P\left(R_T \geq E[R_T] + O^*(\sqrt{T})\right)
$$

$$
\leq O^*(\sqrt{T}) + C_0 \cdot \log(T) \cdot T \cdot P\left(R_T \geq O^*(\sqrt{T})\right)
$$

$$
= O^*(\sqrt{T}) + C_0 \cdot \log(T) \cdot T (1 - P_1) = O^*(\sqrt{T}).
$$

Combining all these together we obtain $E[R_T] = O^*(\sqrt{T})$ which concludes the proof. $\qquad \square$

# E    PROOF OF RESULTS IN SECTION 3.2

## E.1    PROOF OF THEOREM 8

*Proof.* Since the rewards can be unbounded in our setting, we consider truncating the reward with any $\Delta > 0$ for any arm $i$ by $r_i^t = \bar{r}_i^t + \hat{r}_i^t$ where

$$\bar{r}_i^t = r_i^t \cdot \mathbb{1}_{(-\Delta \leq r_i^t \leq \Delta)}, \hat{r}_i^t = r_i^t \cdot \mathbb{1}_{(|r_i^t| > \Delta)}.$$

Then for any parameter $0 < \eta < 1$, we choose such $\Delta$ that satisfies

$$P(r_i^t = \bar{r}_i^t, i \leq K) = P(-\Delta \leq r_1^t \leq \Delta, \ldots, -\Delta \leq r_K^t \leq \Delta)$$

$$= \int_{-\Delta}^{\Delta} \int_{-\Delta}^{\Delta} \ldots \int_{-\Delta}^{\Delta} f(x_1, \ldots, x_K) dx_1 \ldots dx_K \geq 1 - \eta. \tag{23}$$

The existence of such $\Delta = \Delta(\eta)$ follows from elementary calculus.

Let $A = \{|r_i^t| \leq \Delta \text{ for every } i \leq K, t \leq T\}$. Then the probability of this event is

$$P(A) = P(r_i^t = \bar{r}_i^t, i \leq K, t \leq T) \geq (1 - \eta)^T.$$

With probability $(1 - \eta)^T$, the rewards of the player are bounded in $[-\Delta, \Delta]$ throughout the game. Then $R_T^B = \sum_{t=1}^T (\max_i \bar{r}_i^t - \bar{r}_t^i) \leq T \cdot \Delta - \sum_{t=1}^T r_t$ is the regret under event $A$, i.e. $R_T = R_T^B$ with probability $(1 - \eta)^T$. For the EXP3.P algorithm and $R_T^B$ with rewards $\bar{r}_j^t = \frac{\bar{r}_j^t + \Delta}{\Delta}$ satisfying $0 < \bar{r}_j^t < 1$, for every $\delta > 0$, according to **(author?)** (3) we have

$$R_T^B \leq 4\Delta \left( \sqrt{KT \log(\frac{KT}{\delta})} + 4\sqrt{\frac{5}{3}KT \log K} + 8 \log(\frac{KT}{\delta}) \right) \text{ with probability } 1 - \delta.$$

Then we have

$$R_T \leq 4\Delta(\eta) \left( \sqrt{KT \log(\frac{KT}{\delta})} + 4\sqrt{\frac{5}{3}KT \log K} + 8 \log(\frac{KT}{\delta}) \right) \text{ with probability } (1-\delta) \cdot (1-\eta)^T.$$

$\square$

## E.2    PROOF OF THEOREM 9

**Lemma 8.** *For any non-decreasing differentiable function $\Delta = \Delta(T) > 0$ satisfying*

$$\lim_{T \to \infty} \frac{\Delta(T)^2}{\log(T)} = \infty, \qquad \lim_{T \to \infty} \Delta'(T) \leq C_0 < \infty,$$

*and any $0 < \delta < 1, a > 2$ we have*

$$P\left( R_T \leq \Delta(T) \cdot \log(1/\delta) \cdot O^*(\sqrt{T}) \right) \geq (1 - \delta) \left( 1 - \frac{1}{T^a} \right)^T$$

*for any $T$ large enough.*

*Proof.* Let $a > 2$ and let us denote

$$F(y) = \int_{-y}^{y} f(x_1, x_2, \ldots, x_K) dx_1 dx_2 \ldots dx_K,$$

$$\zeta(T) = F\left(\Delta(T) \cdot \mathbf{1}\right) - \left( 1 - \frac{1}{T^a} \right)$$

for $y \in \mathbb{R}^K$ and $\mathbf{1} = (1, \ldots, 1) \in \mathbb{R}^K$. Let also $y_{-i} = (y_1, \ldots, y_{i-1}, y_{i+1}, \ldots, y_K)$ and $x|_{x_i=y} = (x_1, \ldots, x_{i-1}, y, x_{i+1}, \ldots, x_K)$. We have $\lim_{T \to \infty} \zeta(T) = 0$.

The gradient of $F$ can be estimated as

$$\nabla F \leq \left( \int_{-y_{-1}}^{y_{-1}} f\left(x|_{x_1=y_1}\right) dx_2 \dots dx_K, \dots, \int_{-y_{-K}}^{y_{-K}} f\left(x|_{x_K=y_K}\right) dx_1 \dots dx_{K-1} \right).$$

According to the chain rule and since $\Delta'(T) \geq 0$, we have

$$\frac{dF(\Delta(T) \cdot \mathbf{1})}{dT} \leq \int_{-\Delta(T) \cdot \mathbf{1}_{-1}}^{\Delta(T) \cdot \mathbf{1}_{-1}} f\left(x|_{x_1=\Delta(T)}\right) dx_2 \dots dx_K \cdot \Delta'(T) +$$

$$\dots + \int_{-\Delta(T) \cdot \mathbf{1}_{-K}}^{\Delta(T) \cdot \mathbf{1}_{-K}} f\left(x|_{x_K=\Delta(T)}\right) dx_1 \dots dx_{K-1} \cdot \Delta'(T).$$

Next we consider

$$\int_{-\Delta(T)\mathbf{1}_{-i}}^{\Delta(T)\mathbf{1}_{-i}} f\left(x|_{x_i=\Delta(T)}\right) dx_1 \dots dx_{i-1} dx_{i+1} \dots dx_K$$

$$= e^{-\frac{1}{2}a_{ii}(\Delta(T))^2 + \mu_i \Delta(T)} \cdot \int_{-\Delta(T)\mathbf{1}_{-i}}^{\Delta(T)\mathbf{1}_{-i}} e^{g(x_{-i})} dx_1 \dots dx_{i-1} dx_{i+1} \dots dx_K.$$

Here $e^{g(x_{-i})}$ is the conditional density function given $x_i = \Delta(T)$ and thus $\int_{-\Delta(T)\mathbf{1}_{-i}}^{\Delta(T)\mathbf{1}_{-i}} e^{g(x_{-i})} dx_1 \dots dx_{i-1} dx_{i+1} \dots dx_K \leq 1$. We have

$$\int_{-\Delta(T)\mathbf{1}_{-i}}^{\Delta(T)\mathbf{1}_{-i}} f\left(x|_{x_i=\Delta(T)}\right) dx_1 \dots dx_{i-1} dx_{i+1} \dots dx_K$$

$$\leq e^{-\frac{1}{2}a_{ii}(\Delta(T))^2 + \mu_i \Delta(T)}$$

$$\leq e^{-\frac{1}{2}\min_j a_{jj}(\Delta(T))^2 + \max_j \mu_j \Delta(T)}.$$

Then for $T \geq T_0$ we have $\Delta'_T \leq C_0 + 1$ and in turn

$$\zeta'(T) \leq (C_0 + 1) \cdot K \cdot e^{-\frac{1}{2}\min_j a_{jj}(\Delta(T))^2 + \max_j \mu_j \Delta(T)} - a \cdot T^{-a-1}.$$

Since we only consider non-degenerate Gaussian bandits with $\min a_{ii} > 0$, $\mu_i$ are constants and $\Delta(T) \to \infty$ as $T \to \infty$ according to the assumptions in Lemma 8, there exits $C_1 > 0$ and $T_1$ such that

$$e^{-\frac{1}{2}\min_j a_{jj}(\Delta(T))^2 + \max_j \mu_j \Delta(T)} \leq e^{-C_1 \Delta(T)^2} \text{ for every } T > T_1.$$

Since $\lim_{T \to \infty} \frac{\Delta(T)^2}{\log(T)} = \infty$, we have

$$\Delta(T)^2 > \frac{2(a+1)}{C_1} \cdot \log(T) \text{ for } T > T_2.$$

These give us that

$$\zeta(T)' \leq (C_0 + 1)K e^{-2(a+1)\log T} - aT^{-a-1}$$

$$= (C_0 + 1)K e^{-2(a+1)\log T} - ae^{-(a+1)\log T}$$

$$< 0 \text{ for } T \geq T_3 \geq \max(T_0, T_1, T_2).$$

This concludes that $\zeta'(T) < 0$ for $T \geq T_3$. We also have $\lim_{T \to \infty} \zeta(T) = 0$ according to the assumptions. Therefore, we finally arrive at $\zeta(T) > 0$ for $T \geq T_3$. This is equivalent to

$$\int_{-\Delta(T) \cdot \mathbf{1}}^{\Delta(T) \cdot \mathbf{1}} f(x_1, \dots, x_K) dx_1 \dots dx_K \geq 1 - \frac{1}{T^a},$$

i.e. the rewards are bounded by $\Delta(T)$ with probability $1 - \frac{1}{T^a}$. Then by the same argument for $T$ large enough as in the proof of Theorem 4, we have

$$P\left(R_T \leq \Delta(T) \cdot \log(1/\delta) \cdot O^*(\sqrt{T})\right) \geq (1 - \delta)(1 - \frac{1}{T^a})^T.$$

$\square$

**Proof of Theorem 6.** In Lemma 8, we choose $\Delta(T) = \log(T)$, which meets all of the assumptions. The result now follows from $\log T \cdot O^*(\sqrt{T}) = O^*(\sqrt{T})$, Lemma 8 and Theorem 5. $\qquad\square$

### E.3 Proof of Theorem 10

We again utilize the 3 known lemmas, Lemma 5, Lemma 5 and Lemma 7.

**Proof of Theorem 7.** Let us define $F = \{f_j : x \to x_j | j = 1, 2, \ldots, K\}$. Let $x_t = (r_1^t, r_2^t, \ldots, r_K^t)$ where $r_i^t$ is the reward of arm $i$ at step $t$ and let $a_t$ be the arm selected at time $t$ by EXP3.P. Then for any $f_j \in F$, $f_j(x_t) = r_j^t$. In Gaussian-MAB, $\{x_1, x_2, \ldots, x_T\}$ are i.i.d. random variables since the Gaussian distribution $\mathcal{N}(\mu, \Sigma)$ is invariant to time and independent of time. Then by Lemma 5, we have

$$E\left[\max_i \left| \mu_i - \frac{1}{T} \sum_{t=1}^T r_i^t \right|\right] \le 2R_T^c(F).$$

We consider

$$
\begin{aligned}
E\left[|R_T' - R_T|\right] &= E\left[\left| T \cdot \max_i \mu_i - \sum_{t=1}^T \mu_{a_t} - \left( \max_i \sum_{t=1}^T r_i^t - \sum_{t=1}^T r_{a_t}^t \right) \right|\right] \\
&= E\left[\left| T \cdot \max_i \mu_i - \max_i \sum_{t=1}^T r_i^t - \left( \sum_{t=1}^T \mu_{a_t} - \sum_{t=1}^T r_{a_t}^t \right) \right|\right] \\
&\le E\left[\left| T \cdot \max_i \mu_i - \max_i \sum_{t=1}^T r_i^t \right|\right] + E\left[\left| \sum_{t=1}^T \mu_{a_t} - \sum_{t=1}^T r_{a_t}^t \right|\right] \\
&\le E\left[\max_i \left| T \cdot \mu_i - \sum_{t=1}^T r_i^t \right|\right] + E\left[\left| \sum_{t=1}^T \mu_{a_t} - \sum_{t=1}^T r_{a_t}^t \right|\right] \\
&\le 2T R_T^c(F) + 2T_1 R_{T_1}^c(F) + \cdots + 2T_K R_{T_K}^c(F)
\end{aligned}
\tag{24}
$$

where $T_i$ is the number of pulls of arm $i$. Clearly $T_1 + T_2 + \ldots + T_K = T$. By Lemma 6 with $A = F$ we get

$$
\begin{aligned}
R_T^c(F) &= E\left[ \hat{R}_T^c(F) \right] \le E[R(F,T)] \cdot \frac{\sqrt{2 \log K}}{T}, \\
R_{T_i}^c(F) &\le E\left[R(F, T_i)\right] \cdot \frac{\sqrt{2 \log K}}{T_i} \qquad i = \{1, 2, \ldots, K\}.
\end{aligned}
$$

Since $R(F, T)$ is increasing in $T$ and $T_i \le T$, we have $R_{T_i}^c(F) \le E\left[R(F, T)\right] \cdot \frac{\sqrt{2 \log K}}{T_i}$.

We next bound the expected deviation $E\left[|R_T' - R_T|\right]$ based on (24) as follows

$$
\begin{aligned}
E\left[|R_T' - R_T|\right] &\le 2T E[R(F,T)] \frac{\sqrt{2 \log K}}{T} + \sum_{i=1}^K \left[ 2T_i E[R(F,T)] \frac{\sqrt{2 \log K}}{T_i} \right] \\
&\le 2(K+1) \sqrt{2 \log K} E[R(F,T)].
\end{aligned}
\tag{25}
$$

Regarding $E[R(F,T)]$, we have

$$
\begin{aligned}
E[R(F,T)] &= E\left[ \sup_{f \in F} \left( \sum_{t=1}^T f(x_t) \right)^{\frac{1}{2}} \right] = E\left[ \sup_i \left( \sum_{t=1}^T (r_i^t)^2 \right)^{\frac{1}{2}} \right] \\
&\le E\left[ \sum_{i=1}^K \left( \sum_{t=1}^T (r_i^t)^2 \right)^{\frac{1}{2}} \right] \le \sum_{i=1}^K E\left[ \left( T \cdot \max_{1 \le t \le T} (r_t^i)^2 \right)^{\frac{1}{2}} \right] \\
&= \sqrt{T} \cdot \sum_{i=1}^K E\left[ \max_{1 \le t \le T} |r_i^t| \right].
\end{aligned}
\tag{26}
$$

We next use Lemma 7 for any arm $i$. To this end let $Y_t = r_i^t$. Since $x_t$ are Gaussian, the marginals $Y_t$ are also Gaussian with mean $\mu_i$ and standard deviation of $a_{ii}$. Combining this with the fact that a Gaussian random variable is also $\sigma^2$-sub-Gaussian justifies the use of the lemma. Thus $E\left[\max_{1 \le j \le T} |r_i^j|\right] \le a_{i,i} \cdot \sqrt{2 \log T} + \frac{4a_{i,i}}{\sqrt{2 \log T}}$.

Continuing with equation 26 we further obtain

$$E[R(F,T)] \le \sqrt{T} \cdot K \cdot \max_i \left( a_{i,i} \sqrt{2 \log T} + \frac{4a_{i,i}}{\sqrt{2 \log T}} \right)$$

$$= \left( K\sqrt{2T \log T} + \frac{4\sqrt{T}}{\sqrt{2 \log T}} \right) \cdot \max_i a_{i,i}. \tag{27}$$

By combining equation 25 and equation 27 we conclude

$$E\left[|R_T' - R_T|\right] \le 2(K+1)\sqrt{2 \log K} \cdot \max_i a_{i,i} \cdot \left( K\sqrt{2T \log T} + \frac{4\sqrt{T}}{\sqrt{2 \log T}} \right) \tag{28}$$

$$= O^*(\sqrt{T}).$$

We now turn our attention to the expectation of regret $E[R_T]$. It can be written as

$$E[R_T] = E\left[R_T \mathbb{1}_{R_T \le O^*(\sqrt{T})}\right] + E\left[R_T \mathbb{1}_{R_T > O^*(\sqrt{T})}\right]$$

$$\le O^*(\sqrt{T})P\left(R_T \le O^*(\sqrt{T})\right) + E\left[R_T \mathbb{1}_{R_T > O^*(\sqrt{T})}\right] \le O^*(\sqrt{T}) + E\left[R_T \mathbb{1}_{R_T > O^*(\sqrt{T})}\right]$$

$$= O^*(\sqrt{T}) + E\left[R_T \mathbb{1}_{O^*(\sqrt{T}) < R_T < O^*(\sqrt{T}) + E[R_T]}\right] + E\left[R_T \mathbb{1}_{R_T \ge O^*(\sqrt{T}) + E[R_T]}\right]. \tag{29}$$

We consider $\delta = 1/\sqrt{T}$ and $\eta = T^{-a}$ for $a > 2$. We have

$$\lim_{T \to \infty} (1-\delta)(1-\eta)^T = \lim_{T \to \infty} (1-\delta)(1 - \frac{1}{T^a})^T$$

$$= \lim_{T \to \infty} (1-\delta)(1 - \frac{1}{T^a})^{(T^a) \cdot \frac{T}{T^a}} = \lim_{T \to \infty} e^{\frac{T}{T^a}}$$

and

$$\lim_{T \to \infty} \left(1 - (1-\delta)(1-\eta)^T\right) \cdot \log T \cdot T = \lim_{T \to \infty} (1 - e^{\frac{T}{T^a}}) \cdot \log(T) \cdot T \tag{30}$$

$$\le \lim_{T \to \infty} \log(T) \cdot T \cdot T^{1-a} = \lim_{T \to \infty} T^{2-a} \cdot \log(T) = 0.$$

Let $P_1 = P\left(R_T \le \log(1/\delta)O^*(\sqrt{T})\right)$ which equals to $P\left(R_T \le O^*(\sqrt{T})\right)$ since $\log(1/\delta) = \log(\sqrt{T}) = O^*(\sqrt{T})$. By Theorem 6 we have $P_1 = (1-\delta) \cdot (1-\eta)^T$.

Note that $E[R_T] \le C_0 \log(T) \cdot T$ as shown by

$$E[R_T] = E\left[\max_i \sum_{t=1}^T r_i^t - \sum_{t=1}^T r_{a_t}^t\right] \le 2E\left[\max_i \sum_{t=1}^T |r_i^t|\right] \le 2T \cdot E\left[\max_i \max_t |r_i^t|\right]$$

$$\le 2T \cdot \sum_{i=1}^K E\left[\max_t |r_i^t|\right] \le 2T \cdot \sum_{i=1}^K \left( a_{i,i}\sqrt{2 \log T} + \frac{4a_{i,i}}{\sqrt{\log T}} \right)$$

$$\le 2T \cdot \sum_{i=1}^K \max_i a_{i,i} \left( \sqrt{2 \log T} + \frac{4}{\sqrt{\log T}} \right)$$

$$\le C_0 \cdot T \cdot \log(T)$$

for a constant $C_0$.

The asymptotic behavior of the second term in equation 29 reads

$$E\left[R_T \mathbb{1}_{O^*(\sqrt{T}) < R_T < O^*(\sqrt{T}) + E[R_T]}\right] = E\left[R_T \mathbb{1}_{R_T - O^*(\sqrt{T}) \in (0, E[R_T])}\right]$$

$$= E\left[\left(R_T - O^*(\sqrt{T})\right) \mathbb{1}_{R_T - O^*(\sqrt{T}) \in (0, E[R_T])}\right] + O^*(\sqrt{T})$$

$$\leq E\left[R_T\right] P\left(R_T - O^*(\sqrt{T}) \in (0, E\left[R_T\right])\right) + O^*(\sqrt{T})$$

$$\leq E\left[R_T\right] P\left(R_T - O^*(\sqrt{T}) > 0\right) + O^*(\sqrt{T})$$

$$\leq C_0 \log(T) \cdot T \cdot (1 - P_1) + O^*(\sqrt{T}) = O^*(\sqrt{T})$$

where at the end we use equation 30.

Regarding the third term in equation 29, we note that $R'_T \leq E[R_T]$ by the Jensen's inequality. By using equation 28 and again equation 30 we obtain

$$E\left[R_T \mathbb{1}_{R_T \geq O^*(\sqrt{T}) + E[R_T]}\right]$$

$$= E\left[(R_T - R'_T) \mathbb{1}_{(R_T - E[R_T]) \geq O^*(\sqrt{T})}\right] + E\left[R'_T \mathbb{1}_{(R_T - E[R_T]) \geq O^*(\sqrt{T})}\right]$$

$$\leq E\left[|R_T - R'_T|\right] + R'_T \cdot P\left(R_T \geq E\left[R_T\right] + O^*(\sqrt{T})\right)$$

$$\leq E\left[|R_T - R'_T|\right] + E\left[R_T\right] \cdot P\left(R_T \geq E\left[R_T\right] + O^*(\sqrt{T})\right)$$

$$\leq O^*(\sqrt{T}) + C_0 \cdot \log(T) \cdot T \cdot P\left(R_T \geq O^*(\sqrt{T})\right)$$

$$= O^*(\sqrt{T}) + C_0 \cdot \log(T) \cdot T (1 - P_1) = O^*(\sqrt{T}).$$

Combining all these together we obtain $E[R_T] = O^*(\sqrt{T})$ which concludes the proof. $\square$

## F  PROOF OF RESULTS IN SECTION 3.3

For brevity, we define $n = T - 1$.

We start by showing the following proposition that is used in the proofs.

**Proposition 1.** *Let $G(q, \mu), q,$ and $\mu$ be defined as in Theorem 6. Then for any $q \geq 1/3$, there exists a $\mu$ that satisfies the constraint $G(q, \mu) < q$.*

*Proof.* Let us denote $G_1 = \int |qf_0(x) - (1-q)f_1(x)|\, dx, G_2 = \int |(1-q)f_0(x) - qf_1(x)|\, dx.$ Then we have

$$G_1(q, \mu) = \int |qf_0(x) - (1-q)f_1(x)|\, dx$$

$$= \int (qf_0(x) - (1-q)f_1(x)) \mathbb{1}_{qf_0(x) > (1-q)f_1(x)} dx$$

$$+ \int (-qf_0(x) + (1-q)f_1(x)) \mathbb{1}_{qf_0(x) < (1-q)f_1(x)} dx$$

$$= \int (qf_0(x) - (1-q)f_1(x)) \mathbb{1}_{x < g(\mu)} dx + \int (-qf_0(x) + (1-q)f_1(x)) \mathbb{1}_{x > g(\mu)} dx$$

$$= \frac{1}{\sqrt{2\pi}}\left[\int_{-\infty}^{g(\mu)}\left(qe^{-\frac{x^2}{2}} - (1-q)e^{-\frac{(x-\mu)^2}{2}}\right) dx + \int_{g(\mu)}^{\infty}\left(-qe^{-\frac{x^2}{2}} + (1-q)e^{-\frac{(x-\mu)^2}{2}}\right) dx\right]$$

$$= \frac{1}{\sqrt{2\pi}}\left[q\int_{-g(\mu)}^{g(\mu)} e^{-\frac{x^2}{2}} - (1-q)\int_{-g(\mu)+\mu}^{g(\mu)-\mu} e^{-\frac{x^2}{2}}\right]$$

where $g(\mu) = \frac{1}{2}\cdot\mu - \frac{\log(\frac{1-q}{q})}{\mu}$. Similarly we get

$$G_2(q, \mu) = \frac{1}{\sqrt{2\pi}}\left[(1-q)\int_{-g(\mu)}^{g(\mu)} e^{-\frac{x^2}{2}} - q\int_{-g(\mu)+\mu}^{g(\mu)-\mu} e^{-\frac{x^2}{2}}\right].$$

It is easy to establish continuity of $G_1(q, \mu)$ and $G_2(q, \mu)$ on $[0, \infty)$, as well as the continuity of $G(q, \mu)$. Indeed, we have

$$G(q, \mu) = \begin{cases} |1 - 2q| & \mu = 0 \\ \max(q, 1 - q) & \mu \to \infty. \end{cases}$$

Since $q \geq \frac{1}{3}$, then $|1 - 2q| < q$. From continuity of $G(q, \mu)$, there exists $\mu_0 > 0$ such that $G(q, \mu) < q$ for any $\mu \leq \mu_0$. $\qquad\square$

*Proof of Theorem 11.* As in Assumption 1, let the inferior arm set be $I$ and the superior one be $S$, respectively, $P(I) = q$ and $P(S) = 1 - q$. Arms in $I$ follow $f_0(x) = \mathcal{N}(0, 1)$ and arms in $S$ follow $f_1(x) = \mathcal{N}(\mu, 1)$ where $\mu > 0$. According to Assumption 1, at the first step the player pulls an arm from either $I$ or $S$ and receives reward $y_1$. At time step $i > 1$, the reward is $y_i$ and let $b_i$ represent a policy of the player. We can always define $b_i$ as

$$b_i = \begin{cases} 1 & \text{if the chosen arm at step } i \text{ is not in the same arm set as the initial arm,} \\ 0 & \text{otherwise.} \end{cases}$$

Let $a_i \in \{0, 1\}$ be the actual arm played at step $i$. It suffices to only specify $a_i$ is in arm set $I$ ($a_i = 0$) or $S$ ($a_i = 1$) since the arms in $I$ and $S$ are identical. The connection between $a_i$ and $b_i$ is explicitly given by $b_i = |a_i - a_1|$. By Assumption 1, it is easy to argue that $b_i = S_i'(y_1, y_2, ..., y_{i-1})$ for a set of functions $S_2', S_3', \dots, S_n', S_{n+1}'$. We proceed with the following lemma.

**Lemma 9.** *Let the rewards of the arms in set $I$ follow any $L_1$ distribution $f_0(x)$ and in set $S$ follow any $L_1$ distribution $f_1(x)$ where the means satisfy $\mu(f_1) > \mu(f_0)$. Let $B$ be the number of arms played in the game in set $S$. Let us assume the player meets Assumption 1. Then no matter what strategy the player takes, we have*

$$\left| \frac{E[B] - (1-q) \cdot (n+1)}{n+1} \right| \leq \epsilon$$

*where $\epsilon, T, f_0, f_1$ satisfy*

$$G(q, f_0, f_1) + (1 - q)(n - 1) \int |f_0(x) - f_1(x)| \leq \epsilon,$$

$$G(q, f_0, f_1) = \max \left\{ \int |q f_0(x) - (1 - q) f_1(x)| \, dx, \int |(1 - q) f_0(x) - q f_1(x)| \, dx \right\}.$$

*Proof.* We have

$$E[B] = \int (a_1 + a_2 + \cdots + a_{n+1}) f_{a_1}(y_1) f_{a_2}(y_2) \dots f_{a_n}(y_n) \, dy_1 dy_2 \dots dy_n.$$

If $a_1 = 0$, then $a_i = b_i$ and

$$E[B|a_1 = 0] = \int (0 + b_2(y_{1:1}) + \dots + b_{n+1}(y_{1:n})) f_0(y_1) f_{b_2}(y_2) \dots f_{b_n}(y_n) \, dy_1 dy_2 \dots dy_n.$$

If $a_1 = 1$, then $1 - a_i = b_i$ and

$$E[B|a_1 = 1] = \int (1 + 1 - b_2(y_{1:1}) + \cdots + 1 - b_{n+1}(y_{1:n})) f_1(y_1) \dots f_{1-b_n}(y_n) \, dy_1 dy_2 \dots dy_n.$$

This gives us

$$E[B] = q \cdot E[B|a_1 = 0] + (1 - q) \cdot E[B|a_1 = 1]$$
$$= (1 - q)(n + 1)$$
$$\quad + \int (b_2 + \cdots + b_{n+1}) \cdot (q \cdot f_0(y_1) \dots f_{b_n}(y_n) - (1 - q) \cdot f_1(y_1) \dots f_{1-b_n}(y_n)) \, dy_1 dy_2 \dots dy_n.$$

By defining $b_1 = 0$, we have

$$E[B] = (1 - q) \cdot (n + 1) +$$
$$\int (b_2 + \cdots + b_{n+1}) (q \cdot f_{b_1}(y_1) \dots f_{b_n}(y_n) - (1 - q) \cdot f_{1-b_1}(y_1) \dots f_{1-b_n}(y_n)) \, dy_1 dy_2 \dots dy_n.$$

For any $1 \leq m \leq n$ we also derive

$$\int \left| \prod_{i=1}^{m} f_{b_i}(y_i) - \prod_{i=1}^{m} f_{1-b_i}(y_i) \right| dy_1 dy_2 \ldots dy_m$$

$$\leq \int \prod_{i=1}^{m-1} f_{b_i}(y_i) \left| f_{b_m}(y_m) - f_{1-b_m}(y_n) \right| dy_1 dy_2 \ldots dy_m +$$

$$\int \left| \prod_{i=1}^{m-1} f_{b_i}(y_i) - \prod_{i=1}^{m-1} f_{1-b_i}(y_i) \right| f_{1-b_m}(y_m) \, dy_1 dy_2 \ldots dy_m$$

$$\leq \int |f_0(x) - f_1(x)| \, dx + \int \left| \prod_{i=1}^{m-1} f_{b_i}(y_i) - \prod_{i=1}^{m-1} f_{1-b_i}(y_i) \right| f_{1-b_m}(y_m) \, dy_1 dy_2 \ldots dy_m \quad (31)$$

$$= \int |f_0(x) - f_1(x)| \, dx + \int \left| \prod_{i=1}^{m-1} f_{b_i}(y_i) - \prod_{i=1}^{m-1} f_{1-b_i}(y_i) \right| dy_1 dy_2 \ldots dy_{m-1}$$

$$\leq 2 \cdot \int |f_0(x) - f_1(x)| \, dx + \int \left| \prod_{i=1}^{m-2} f_{b_i}(y_i) - \prod_{i=1}^{m-2} f_{1-b_i}(y_i) \right| dy_1 dy_2 \ldots dy_{m-2}$$

$$\leq m \int |f_0(x) - f_1(x)| \, .$$

This provides

$$\left| \frac{E[B] - (1-q) \cdot (n+1)}{n+1} \right|$$

$$\leq \int \left| q \cdot \prod_{i=1}^{n} f_{b_i}(y_i) - (1-q) \cdot \prod_{i=1}^{n} f_{1-b_i}(y_i) \right| dy_1 dy_2 \ldots dy_n$$

$$\leq \int \prod_{i=1}^{n-1} f_{b_i}(y_i) \left| q \cdot f_{b_n}(y_n) - (1-q) \cdot f_{1-b_n}(y_n) \right| dy_1 dy_2 \ldots dy_n +$$

$$\int \left| (1-q) \cdot \prod_{i=1}^{n-1} f_{b_i}(y_i) - (1-q) \cdot \prod_{i=1}^{n-1} f_{1-b_i}(y_i) \right| f_{1-b_n}(y_n) \, dy_1 dy_2 \ldots dy_n$$

$$\leq \max \left\{ \int |q \cdot f_0(x) - (1-q) \cdot f_1(x)| \, dx, \int |(1-q) \cdot f_0(x) - q \cdot f_1(x)| \, dx \right\} +$$

$$(1-q) \cdot \int \left| \prod_{i=1}^{n-1} f_{b_i}(y_i) - \prod_{i=1}^{n-1} f_{1-b_i}(y_i) \right| dy_1 dy_2 \ldots dy_{n-1}$$

$$\leq \max \left\{ \int |q \cdot f_0(x) - (1-q) \cdot f_1(x)| \, dx, \int |(1-q) \cdot f_0(x) - q \cdot f_1(x)| \, dx \right\} +$$

$$(1-q) \cdot (n-1) \cdot \int |f_0(x) - f_1(x)| \, ,$$

where the last inequality follows from (31). The statement of the lemma now follows. $\qquad \square$

According to Proposition 1, there is such $\mu$ satisfying the constraint $G(q, \mu) < q$. Note that $G(q, \mu) = G(q, f_0, f_1)$. Then we can choose $\epsilon$ to be any quantity such that $G(q, \mu) < \epsilon < q$. Finally, there is $T$ satisfying $T \leq \frac{\epsilon - G(q, \mu)}{(1-q) \cdot \int |f_0(x) - f_1(x)|} + 2$ that gives us

$$G(q, \mu) + (1-q)(T-2) \int |f_0(x) - f_1(x)| \leq \epsilon.$$

By choosing $\epsilon, T, \mu$ as above, by Lemma 9 we have

$$\left| \frac{E[B] - (1-q) \cdot T}{T} \right| < \epsilon,$$

which is equivalent to $E[B] < (1 - q + \epsilon) \cdot T$. Therefore, regret $R_T'$ satisfies, with $A$ being the number of arm pulls from $I$, inequality

$$R_T' = \sum_t \max_k(\mu_k) - \sum_t E[y_t] = T\mu - \sum_t E[y_t] = T\mu - (E[B] \cdot \mu + E[A] \cdot 0)$$

$$\geq T\mu - (1 - q + \epsilon)\mu T = (q - \epsilon)\mu T.$$

This yields $R_T^L = \inf \sup R_T' \geq (q - \epsilon) \cdot \mu T.$ □

Theorem 12 follows from Theorem 11 and Proposition 1.

*Proof of Theorem 13.* The assumption here is the special case of Assumption 1 where there are two arms and $q = 1/2$. Set $I$ follows $f_0$ and $S$ follows $f_1$ where $\mu(f_0) < \mu(f_1)$.

In the same was as in the proof of Theorem 8 we obtain

$$R_L(T) \geq \left(\tfrac{1}{2} - \epsilon\right) \cdot T \cdot \mu$$

under the constraint that $n/2 \cdot \int |f_0 - f_1| = n/2 \cdot \text{TV}(f_0, f_1) < \epsilon$ where TV stands for total variation. Here we use $G(1/2, \mu) = 1/2 \cdot \text{TV}(f_0, f_1)$. Setting $\epsilon = 1/4$ yields the statement. □

In the Gaussian case it turns out that $\epsilon = 1/4$ yields the highest bound. For total variation of Gaussian variables $N(\mu_1, \sigma_1^2)$ and $N(\mu_2, \sigma_2^2)$, **(author?)** (12) show that

$$\text{TV}\left(\mathcal{N}\left(\mu_1, \sigma_1^2\right), \mathcal{N}\left(\mu_2, \sigma_2^2\right)\right) \leq \frac{3|\sigma_1^2 - \sigma_2^2|}{2\sigma_1^2} + \frac{|\mu_1 - \mu_2|}{2\sigma_1},$$

which in our case yields $TV \leq \frac{\mu}{2}$. From this we obtain $\mu \cdot T \geq \epsilon$ and in turn $R_T^L \geq \epsilon \cdot (\frac{1}{2} - \epsilon)$. The maximum of the right-hand side is obtained at $\epsilon = \frac{1}{4}$. This justifies the choice of $\epsilon$ in the proof of Theorem 1.

# G CONTRIBUTION

Our contributions are two-fold. On the one hand, our optimal regret holds for $T$ being large enough in unbounded bandits. On the other hand, the lower bound regret suggests a lower bound on $T$ to achieve sublinear but not necessarily optimal regret as a by-product. The question for any $T$ points a future direction.

## G.1 UPPER BOUNDS

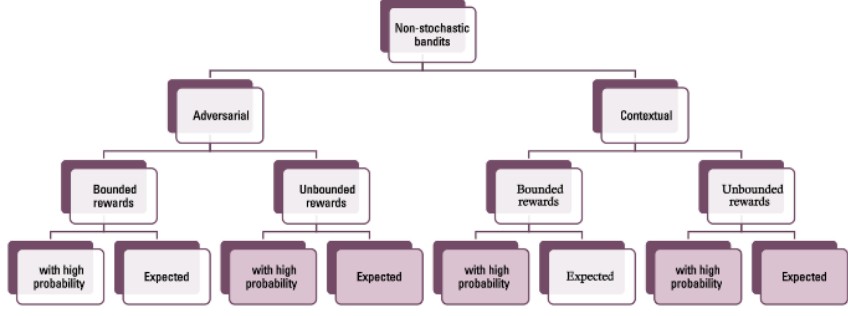

Figure 4: The framework of regret analysis in non-stochastic bandits.

As we can see in Figure 4, the domain of regret analyses for non-stochastic regret bounds can fall into 8 sub-categories by taking all the possible combinations of $A$ and $B$ and $C$, where $A = \{Contextual, Adversarial\}$, $B = \{Bounded\,rewards, Unbounded\,rewards\}$, $C = \{High\,probability\,bound, Expected\,bound\}$ to name a few. The colored boxes in the leaf nodes correspond to the results in this paper and the remained boxes are already covered by the existing literature. For contextual bandits, establishing a high probability regret bound is non-trivial even

for bounded rewards since regret in a contextual setting significantly differs from the one in the adversarial setting. To this end, we propose a brand new algorithm EXP4.P that incorporates EXP3.P in adversarial MAB with EXP4. The analysis for regret of EXP4.P in unbounded cases is quite general and can be extended to EXP3.P without too much effort.

To conclude, the theoretical analyses regarding the upper bound fill the gap between the regret bound in **(author?)** (3) and all others in Figure 4.

### G.2 LOWER BOUNDS

Table 1: Boundary for $T$ as a function of $\mu$

| $\mu$ | $10^{-5}$ | $10^{-4}$ | $10^{-3}$ | $10^{-2}$ | $10^{-1}$ |
|---|---|---|---|---|---|
| Upper bound for $T$ | 25001 | 2501 | 251 | 26 | 3.5 |

In view of unbounded bandits, the previous lower bound in **(author?)** (3) does not hold since unboundedness apparently increases regret. The relationship between the lower bound and time horizon is listed in Table 1 to facilitate the understanding of the lower bound. More precisely, Table 1 provides the values of the relationship between $\mu$ and largest $T$ in the Gaussian case where the inferior arms are distributed based on the standard normal and the superior arms have mean $\mu > 0$ and variance 1. As we can observe in the table, the maximum of $T$ for the lower bound to hold changes with instances. A small $\mu$ means the lower bound on regret of order $T$ holds for larger $T$. For example, there is no way to attain regret lower than $T \cdot 10^{-4}/4$ for any $1 \leq T \leq 2501$. The function decreases very quickly. This coincides with the intuition since it would be difficult to distinguish between the optimal arm and the non-optimal ones given their rewards are close. A lower bound for large $T$ remains open.

