# OpenReview forum: "Regret Bounds and Reinforcement Learning Exploration of EXP-based Algorithms"
_ICLR.cc/2025/Conference — ICLR 2025 Conference Withdrawn Submission_

### Official Review · Reviewer_pF6n · 2024-10-29

**Soundness:** 1
**Presentation:** 1
**Contribution:** 1
**Rating:** 3
**Confidence:** 4

**Summary:**

The paper proposes several variants of EXP algorithms for bandits and RL. It provides high probability regret bounds for different settings, and evaluates the performances on some RL games.

**Strengths:**

MAB/RL with unbounded rewards is a challenging setting.

**Weaknesses:**

- The presentation of this work lacks clarity and requires significant improvement. The problem formulation is unclear, with various assumptions scattered throughout the paper.
- Several theoretical results seem problematic. For example, the high probability results in this work show that the regret bounds hold with probability (1-\delta)(1-\eta)^T. However, as $T$ increases, this probability can become arbitrarily small, which indicates that the bounds do not hold “in high probability” any more.
- Another questionable result is the Corollary in Sec 3.1. It claims that when the horizon becomes 1, EXP4.P and EXP4-RL can become equivalent, thus the results of Theorems 4 and 7 apply to this case.  However, the design of EXP4-RL has a fundamental differences with EXP4.P: EXP4-RL updates the policy chosen by each expert (Q-network) after each episode with the collected data, while that under EXP4.P is not updated by the algorithm. The updating of the experts’ policies introduce complicated interactions between the experts and the decision-making agent, which is not captured in the analysis of EXP4.P, thus the results no longer hold.

**Questions:**

- MAB/RL with sub-gaussian rewards has been well-studied in the literature. What are the new challenges in the problems considered in this work?

---

### Official Review · Reviewer_hMKE · 2024-11-03

**Soundness:** 2
**Presentation:** 2
**Contribution:** 3
**Rating:** 3
**Confidence:** 2

**Summary:**

The paper introduces a novel algorithm, EXP4.P, designed to efficiently explore adversarial contextual bandit problems with unbounded rewards. This algorithm is further extended to EXP4-RL for exploring Markov Decision Processes (MDPs) with unbounded rewards. The authors evaluate EXP4-RL on challenging environments, demonstrating state-of-the-art performance.

**Strengths:**

1. The exploration of environments with unbounded rewards is a crucial and challenging issue in the field.

2. The algorithms are supported by strong theoretical guarantees and comprehensive experimental comparisons.

3. The paper extends the understanding of EXP-type algorithms, offering techniques that may be generalizable to similar algorithms.

**Weaknesses:**

The presentation needs improvement.
- The introduction is too extensive, which may dilute the focus on the core contributions. A more concise introduction with a clear statement of contributions could be better. The comparisons with related works could be separately discussed.
- The main results, particularly Theorems 2 to 4, are not presented clearly. If these theorems share common assumptions, it would be beneficial to outline these assumptions outside the theorems. Consolidating the main results into one or two comprehensive theorems, followed by corollaries for additional insights, could improve clarity.
- In line 786, there are '(authors)?' not addressed yet.

**Questions:**

1. Could the authors offer a more detailed comparison with closely related works? Specifically, it would be helpful to understand the contexts in which unbounded rewards have not been thoroughly studied versus where they have been adequately addressed. For instance, stochastic contextual linear bandits do not typically suffer from issues with sub-Gaussian noise on rewards, as discussed in [1].

2. Could the authors provide an intuitive explanation of why unbounded rewards pose significant difficulties in adversarial settings? And why the modification in EXP4.P is in the spirit of EXP3.P but the terms are much different?

---

### Official Review · Reviewer_UD4P · 2024-11-03

**Soundness:** 2
**Presentation:** 2
**Contribution:** 2
**Rating:** 5
**Confidence:** 2

**Summary:**

This work studies the problem of EXP-based exploration in bandits and RL. A set of theoretical results have been provided, including contextual bandit with bounded reward, linear/stochastic contextual bandit with unbounded reward, MAB with unbounded reward, and also MDP. Experimental results on RL games are also reported to demonstrate the efficiency of the proposed design.

**Strengths:**

- The problem of EXP-based exploration is an interesting theoretical topic worth exploration in both bandits and RL. While I am not exactly familiar with the SOTA studies, this work claims to address several requirements (e.g., bounded rewards) and also new problem settings (e.g., high probability bounds), which I believe is of interests to the community.

- The study is comprehensive in both theory and experiments. Different settings ranging from MAB, contextual bandits and RL are studied. The experimental results are also helpful in providing supporting evidences to the proposed designs.

**Weaknesses:**

My main concern of this work is on its clarity, especially regarding the following aspects:

- The main contribution of this work. This work targets a lot of problems, which is a strength; however, at the same time, I believe the authors can do a better job by explaining the addressed core problems and settings more clearly, instead of providing all the results in the main paper. The discussions in Appendix G are helpful, while I do not think it covers the entire work (e.g., the studies on RL).

- Another suggestion is that probably the presentation can be improved if the notations of the settings, e.g., the different regrets can be defined when they are discussed. The current presentation with all notations flooded in Section 2 is a bit dry in the first place, while hard to refer to upon the discussions.

-  I am in general wondering about the core technical contributions/technical challenges in this work. The authors have provided a full set of results, which I appreciated; however, there are limited discussions on the intuitions. E.g., in EXP4.P, the highlighted terms are the keys in obtaining the desired results I conjecture, whose form (i.e., why they are constructed this way) can be explained a bit more.

**Questions:**

My main suggestions are listed in the weakness section, please find below for a few additional questions:

- In line 256, it claims that the regret in Theorem 1 does not depend on N, while it seems there are several terms that depending on N in the formula, especially the second term that is of order $\sqrt{NT}$?

- In line 226, it claims that the expected regret is bounded using  Rademacher complexity and VC dimension, I would appreciate a pointer to where such usage happens.

---

### Official Review · Reviewer_UkFj · 2024-11-11

**Soundness:** 2
**Presentation:** 1
**Contribution:** 2
**Rating:** 5
**Confidence:** 4

**Summary:**

The paper studies EXP-type algorithms for adversarial bandits and contextual bandits with unbounded rewards. It proposes EXP3.P and EXP4.P algorithms for these settings, and derived corresponding regret bounds. It also proposes an extension of EXP4.P to RL setting and derive regret bound for it. Further numerical experiments are done with EXP4-RL for Mountain Car and Montezuma's revenge to show that EXP4-RL induces better exploratory behaviour than RND (Random Network Distillation).

**Strengths:**

1. Extending EXP type algorithms to unbounded reward settings is interesting.
2. Specifically, its impact for stochastic contextual bandits with unbounded rewards is novel and can be helpful to remove the typical bounded reward assumptions in contextual bandits.
3. Further generalisation of EXP4.P to RL and its numerical validations are useful contributions to design exploration strategies in hard RL problems.

**Weaknesses:**

1. The writing is very hard to decode. It is better
2. The citation style is strange and uninformative. Generally () is for eqref and [] for citation. Also, there are improperly compiled (e.g. (author?) (3)). Please check them for better readability.
3. The simple regret definition used here is a bit strange. (Expected) simple regret is the difference in expected reward obtained by optimal policy and present policy at time $t$ [1,2]. Why calling this new definition "simple regret"?
4. The experimental evaluations have some shortcomings. Please check questions 2-4.

[1] Bubeck, S., Munos, R., and Stoltz, G. (2009). Pure exploration in multi-armed bandits problems. In The 20st International conference on Algorithmic learning theory, pages 23–37. Springer.
[2] Zixin Zhong, Wang Chi Cheung, Vincent Y. F. Tan (2023). Achieving the Pareto Frontier of Regret Minimization and Best Arm Identification in Multi-Armed Bandits. Trans. Mach. Learn. Res.

**Questions:**

1. What is the main technical novelty that allows you to take EXP4 to unbounded reward settings? What was the main technical challenge and which key lemmas allow you to solve it? This is not explicit from the writing but it is the main technical contribution in my view. Can you explicate it?
2. Why choosing (only) RND for experiments? Why not any other and more established baseline for Deep RL, like intrinsic motivation, entropy regularized RL (e.g. SAC)? A more exhaustive experimental evaluation in the deep RL setting would be convincing.
3. Why not demonstrating the cumulative reward in each of the experiments? In the end, the theoretical claims are that making the proposed changes make EXP4 a regret-efficient algorithm in unbounded reward setting. I do not see how this is reflect in experiments.
4. In experiments, the environments studied have only bounded rewards. Can we see some numerical evidence where EXP3 and EXP4 fail while EXP4.P and variants are better performing?
5. How does the result compare with other stochastic contextual bandit algorithms that can handle unbounded rewards, such as [3], [4] etc.
6. A philosophical query is: why studying unbounded rewards in adversarial setting is interesting? Except technical challenges, what does it bring extra with respect to stochastic setting with unbounded rewards or unbounded corruptions?

[3] Marc Abeille and Alessandro Lazaric. Linear thompson sampling revisited. In AISTATS, 2017.
[4] Vecchia, R.D., & Basu, D. (2023). Stochastic Online Instrumental Variable Regression: Regrets for Endogeneity and Bandit Feedback. arXIv 2302.09357.

---

### Note · Authors · 2024-11-27

I have read and agree with the venue's withdrawal policy on behalf of myself and my co-authors.